# A coupled mechano-biochemical model for cell polarity guided anisotropic root growth

**Marco Marconi[1], Marcal Gallemi[2], Eva Benkova[2], Krzysztof Wabnik[1]***

[1]Centro de Biotecnología y Genómica de Plantas, Universidad Politécnica de Madrid (UPM)—Instituto Nacional de Investigación y Tecnología Agraria y Alimentaria (INIA), Pozuelo de Alarcón, Spain; [2]Institute of Science and Technology (IST), Klosterneuburg, Austria

**Abstract** Plants develop new organs to adjust their bodies to dynamic changes in the environment. How independent organs achieve anisotropic shapes and polarities is poorly understood. To address this question, we constructed a mechano-biochemical model for *Arabidopsis* root meristem growth that integrates biologically plausible principles. Computer model simulations demonstrate how differential growth of neighboring tissues results in the initial symmetry-breaking leading to anisotropic root growth. Furthermore, the root growth feeds back on a polar transport network of the growth regulator auxin. Model, predictions are in close agreement with in vivo patterns of anisotropic growth, auxin distribution, and cell polarity, as well as several root phenotypes caused by chemical, mechanical, or genetic perturbations. Our study demonstrates that the combination of tissue mechanics and polar auxin transport organizes anisotropic root growth and cell polarities during organ outgrowth. Therefore, a mobile auxin signal transported through immobile cells drives polarity and growth mechanics to coordinate complex organ development.

**\*For correspondence:**
k.wabnik@upm.es

## Editor's evaluation

The authors have created the first detailed model combining the mechanics of root growth with the dynamic regulation of auxin transport and patterning. Their novel model is capable of explaining the anisotropic longitudinal growth of plant roots and the complicated patterns of polarized auxin transport underlying auxin patterning.

## Introduction

Plants are remarkable organisms because they can successively produce new organs from stem cell reservoirs, and adapt to the dynamic changes in the environment. For example, *Arabidopsis thaliana* roots show coordinated growth involving local interactions between adjacent non-mobile cells to sustain the optimal exploration of soil-derived resources (*O'Brien et al., 2016*), and to provide water and mineral absorption as well as stability on the ground (*Chapman et al., 2012*). The root elongates along the principal direction of growth (referred to as the anisotropy) during the late stages of embryogenesis but the mechanisms underlying the emergence of root growth anisotropy are poorly understood (*Bou Daher et al., 2018*).

Root maturates following a sequence of asymmetric cell divisions and cell expansion that involves the growth regulator auxin (*Adamowski and Friml, 2015*). Auxin is transported in a directional (polar) manner (*Wisniewska et al., 2006*) that requires the polar subcellular localization of the PIN-FORMED (PIN) auxin efflux carriers (*Adamowski and Friml, 2015*; *Benková et al., 2003*). Subsequently, auxin

controls cell elongation through weakening or stiffening of the cell walls depending on threshold concentrations (*Barbez et al., 2017*; *Majda and Robert, 2018*).

In general, cell growth is associated with changes in cytoskeleton components, such as cortical microtubules (CMTs), actin, and cell wall elements (*Adamowski et al., 2019*; *Siegrist and Doe, 2007*). Among those, CMTs integrate mechanical stresses (*Hamant et al., 2019*; *Hamant and Haswell, 2017*) into dynamic cytoskeleton rearrangements, constraining the directional trafficking of membrane proteins, small signaling molecules, and cell wall building components (*Adamowski et al., 2019*; *Siegrist and Doe, 2007*).

This complexity of root growth mechanisms has long attracted theoreticians on the quest to identify its underlying principles. In the last decade, several computational models of root development yielded important insights into auxin-dependent growth and zonation of mature roots (*Morales-Tapia and Cruz-Ramírez, 2016*; *Rutten and Ten Tusscher, 2019*; *Wabnik et al., 2011*). Yet, these models typically incorporate pre-defined patterns of polar auxin flow on idealized geometries (e.g. cell grids) (*Band et al., 2014*; *Grieneisen et al., 2007*; *Mähönen et al., 2014*; *Mironova et al., 2010*; *Wabnik et al., 2013*) and rarely integrate growth mechanics (*De Vos et al., 2014*; *Fozard et al., 2013*; *Jensen and Fozard, 2015*). The major challenge is, however, to identify the elusive mechanisms that generate initial symmetry breaking, leading to anisotropic root growth and the establishment of a sophisticated polar auxin transport network. Despite numerous experimental and modeling studies, this issue remains largely unaddressed.

A comprehensive approach to tackle these challenges should accommodate both biochemical and biomechanical aspects of early organ growth at cellular resolution, but so far this has represented a major challenge in both plant and animal modeling fields (*Delile et al., 2017*; *Fletcher et al., 2014*; *Heisler et al., 2010*).

Here, we address the problem of anisotropic root growth and cell polarity patterning using an advanced computer modeling strategy that combines growth mechanics with biochemical transport at single-cell resolution. Our model is based on a set of biologically plausible principles and is capable of recapitulating the establishment of root meristem anisotropic elongation through auxin-dependent root growth, tissue biomechanics, and polar auxin transport network from the small population of non-polar differentiated cells.

## Results

### Anisotropic root growth results from the differential expansion of neighboring tissues

Plant embryogenesis follows the fertilization of the egg, and successive formation of the zygote (*Park and Harada, 2008*). Initially, the zygote contracts transiently to later elongate, setting the embryonic axis within a few hours; after several cell divisions, the aerial and root parts are already clearly distinguishable (*Kimata et al., 2016*). The stem cell niche is initiated during the 'heart' developmental stage (*ten Hove et al., 2015*).

Therefore, we chose this stage for the construction of the computer model of the root meristem as it provides an ideal starting point for the entire organ establishment, assuming uniform growth and non-polar distribution of auxin transporters and a set of differentiated cells. By digitizing the confocal microscopy images of the heart-stage of *A. thaliana* embryo, we build 2D cellular meshes with Morpho-GraphX (*Barbier de Reuille et al., 2015*; *Figure 1—figure supplement 1*). To model the pure growth mechanics at a single-cell resolution, we used these meshes at the start of each simulation. We implemented the organ growth framework based on Position-based Dynamics (PBD), a modeling technique adapted from computer graphics (*Müller et al., 2007*; *Marconi and Wabnik, 2021a*; *Marconi and Wabnik, 2021b*, *Figure 1—figure supplement 2A*). PBD approximates physical forces using a set of growth constraints (*Müller et al., 2007*). These constraints reproduce internal turgor pressure, visco-elastic behavior of plant cell walls, and mechanical deformation (strain) (*Figure 1—figure supplement 2B*; see Materials and methods section for PBD details). Because these constraints are sequentially projected over the vertices of the 2D meshes by directly updating the position of vertices, the PBD method avoids the slow numerical time integration step used in classical force-based methods (*Müller et al., 2007*). Thus PBD is faster and more stable than other physically based approaches such as mass-spring systems (*Müller et al., 2007*) or finite element methods (FEM)(*Bidhendi and Geitmann*,

*2018*; *Fayant et al., 2010*), and therefore ideally suited for complex organ modeling at cellular resolutions. This numerical stability of PBD is critical when dealing with growing entities and expanding complexities. Another important advantage of using PBD is that this method can be explicitly defined on a single-cell or even subcellular level which remains a major challenge for continuous mechanics FEM-based approaches (*Boudon et al., 2015*).

We explored through mechanical model simulations the potential mechanisms of anisotropic root growth which have remained unknown until now. Recent studies suggest that differential cell growth produced by mechanical interactions may regulate organogenesis independently from genetic control, and potentially feedback on it. Some examples include *Arabidopsis* trichomes emerging from sepals (*Hervieux et al., 2017*) or the tomato shoot apical meristem (*Kierzkowski et al., 2012*). Interestingly, in the hypocotyl, the CMT array is transversely oriented to the hypocotyl growth axis during the elongation phase and longitudinally oriented when elongation stops (*Le et al., 2005*). Despite these observations, we still lack a mechanistic understanding of CMTs, actin, and cell wall component together regulate anisotropy which limits detailed modeling of individual components of the cytoskeleton network. Modeling each of these components separately would require the integration of a large number of biological properties, most of which are poorly understood. To avoid additional complexities, we decided to approximate the outcome of processes involved in cytoskeleton dynamics by an abstract 'anisotropy factor' (AF). In the model, the AF denotes the tendency of the cell to grow anisotropically and can be reoriented through the action of external stimuli or forces. Explicitly, the AF reorientation follows cell deformation, creating a feedback mechanism that further reinforces the anisotropic growth (see Materials and methods for more details, Position-based dynamics implementation).

We then hypothesized that perhaps differential growth of tissues at the root-shoot interface (RSI) during late embryogenesis could potentially trigger initial symmetry breaking, leading to anisotropic root growth (*Figure 1A–B*). To test this hypothesis, we performed model simulations first by assuming uniform growth of root and RSI (*Figure 1A*, *Figure 1—video 1*). In this scenario, we could only observe the strong isotropic growth at the basal part of the embryo (BPE) (*Figure 1A*). Anisotropy-generating elements such as CMTs are typically perpendicular to the maximal growth direction (*Hamant et al., 2019*), yet, the lack of any mechanical growth restriction leads to isotropic deformation. In contrast, faster growth of the BPE compared to the adjacent embryonic tissues yields a strong anisotropic expansion (*Figure 1B* and *Figure 1—video 1*). The plausible explanation for this is that a slowly growing RSI prevents the expansion of the faster-growing BPE in the radial direction; the BPE gradually enlarges longitudinally, generating deformation (strain). Then, this deformation feeds back on the BPE growth, creating the desired anisotropy. We tested these model predictions by quantifying the growth increase over 6 hr in radicle and hypocotyl of young seedlings using time-lapse confocal microscopy imaging (*Figure 1C*; *Zhu et al., 2019*). Indeed, the emerging root radicle grew ~4 x faster than the adjacent hypocotyl (*Figure 1D*) in the initial outgrowth phase which further supports our model.

To further confirm that growth anisotropy indeed emerges from differential growth rates, and not from an existing conflict of growth direction, we quantified the degree of cellular anisotropy in both scenarios and found that anisotropy forms gradually without predominant growth conflicts, but is rather dictated by differential growth at the RSI (*Figure 1—figure supplement 3*). Finally, we tested the robustness of the model by relaxing the differential growth assumption during the late outgrowth stimulation (*Figure 1—figure supplement 4*). After a short period of growth after which the anisotropy is established, the organ is still capable of maintaining anisotropic elongation even if differential growth at the RSI is abolished (*Figure 1—figure supplement 4*). This result further strengthens the notion that the differential growth between adjacent tissues is, in principle, sufficient to generate root growth anisotropy.

In summary, model simulations and experiments jointly suggest that anisotropic root growth results from differential growth rates of neighboring tissues. This oriented growth further restricts root elongation, primarily along the longitudinal axis.

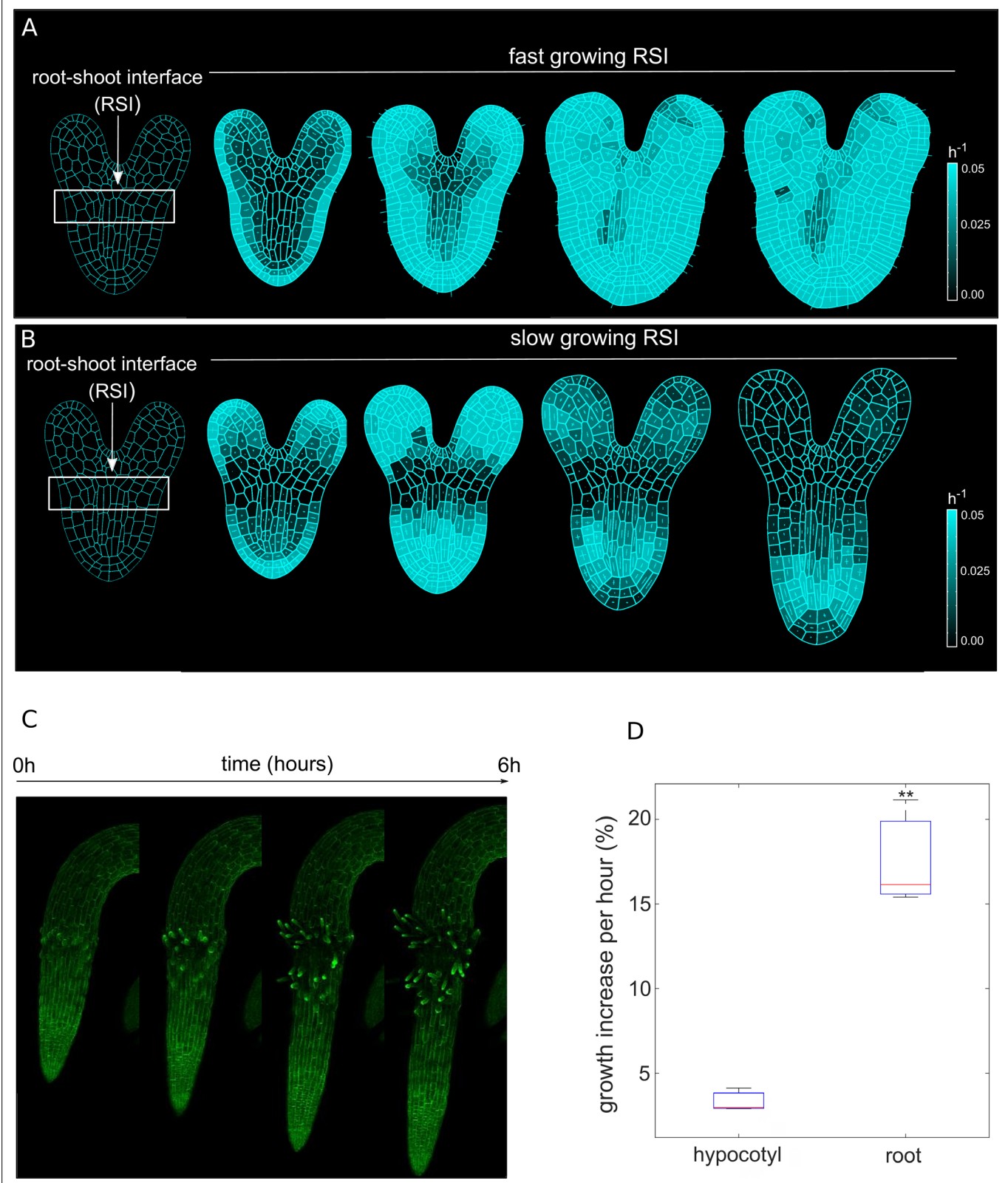

**Figure 1.** Differential cell growth at the RSI produces the emergent anisotropic expansion of the root. (**A, B**) Simulated root growth mechanics with uniform growth rates at the RSI (**A**). The RSI and the root are allowed to grow at the same rate, producing an isotropic growth pattern. (**B**) Simulated root outgrowth by assuming differential growth rate at the RSI. The root grows anisotropically since the growing cell deformation causes the gradual AF stabilization orthogonal to this deformation. All the cells are allowed to grow at the same rate (purely mechanical model). Simulations have been run for

*Figure 1 continued on next page*

*Figure 1 continued*

300 time steps. The white lines represent the principal directions of growth. (**C**) Screenshots from time-lapse imaging of growing radicle with PIP-GFP plasma membrane marker (*Zhu et al., 2019*) for a total time of 6 hr (six time points). (**D**) Size increase per hour (in %) for adjacent organs radicle (root) and hypocotyl quantified from the time-lapse confocal imaging of three independent plants (n = 3). **p-value = 0.0015 a one-way ANOVA with post-hoc Tukey's HSD.

The online version of this article includes the following video and figure supplement(s) for figure 1:

**Source data 1.** Source data used to generate *Figure 1D*.

**Figure supplement 1.** Overview of schematic procedure for the segmentation of microscopy images of an *A*.

**Figure supplement 2.** Position-Based Dynamics schematics (**A**) Comparison between force-based methods (left panel) and PBD (right panel).

**Figure supplement 3.** Effect of differential cell growth at the RSI on emergent cell growth anisotropy.

**Figure supplement 4.** Testing the robustness of the proposed symmetry-breaking model by removing the differential growth between tissues later during the simulation.

**Figure 1—video 1.** Differential cell growth at the embryo RSI produces anisotropic expansion of the root - movie related to Figure 1.
https://elifesciences.org/articles/72132/figures#fig1video1

## Organ growth patterns arise through the interplay between anisotropic growth and polar auxin flow

Our purely mechanical growth model suggests that differential growth at the RSI could trigger anisotropic root growth (*Figure 1*). In *A. thaliana* root, cellular auxin levels play a key role in regulating growth, and auxin levels can be tuned through intercellular transport, involving auxin influx and efflux carriers (*Adamowski and Friml, 2015*). While auxin influx carriers of the AUX/LAX family are typically uniformly distributed around the cell membranes (*Kleine-Vehn and Friml, 2008*), PIN auxin efflux carriers show polar subcellular localization in the root that directs the auxin flow rootward or shootward (*Wisniewska et al., 2006*). Also, PIN proteins are prominent markers of cell polarity that continuously recycle between the plasma membrane and endosomal compartments (*Kleine-Vehn et al., 2011*). The mechanisms underlying PIN trafficking are still poorly understood, however, chemical treatments of actin, microtubules, and cell wall components with disruptive agents suggest the involvement of these cytoskeleton components in the regulation of PIN polar trafficking (*Baskin, 2005*; *Kleine-Vehn et al., 2008*; *Feraru et al., 2011*).

The coexistence of growth polarity and dominant PIN localization in many roots cells suggests that growth anisotropy and PIN polarity may be mechanistically entangled as previously shown for the shoot apical meristem (*Heisler et al., 2010*). Because tissue mechanics control growth anisotropy it is plausible to conceive possible feedback on PIN polarity that modulates deposition of the cell wall and cytoskeleton components (*Braybrook and Peaucelle, 2013*; *Feraru et al., 2011*; *Heisler et al., 2010*). Based on these experimental observations, we thought of a scenario where the AF restricts the axis along which PINs are delivered. This would recreate the correlation between anisotropic growth and PIN localization, but it would not determine the preferential direction (rootward, shootward, or lateral) of auxin flow. Therefore, to define the actual direction of auxin movement in our model other mechanisms of likely biochemical nature are required.

Auxin modulates the trafficking of PIN proteins in a feedback-dependent manner by a yet unknown molecular mechanism (*Adamowski and Friml, 2015*; *Narasimhan et al., 2021*). Several theories for the establishment of PIN polarities have been proposed, i.e. through sensing the net auxin flux through the cell (*Feugier and Iwasa, 2006*; *Mitchison, 1997*; *Rolland-Lagan and Prusinkiewicz, 2005*; *Stoma et al., 2008*), auxin concentrations (*Jönsson et al., 2006*; *Merks et al., 2007*; *Smith et al., 2006*; *Wabnik et al., 2010*), the auxin gradient inside the cell (*Kramer, 2009*) or their combination (*Cieslak et al., 2015*). We tested scenarios of the auxin effect on its PIN-mediated transport using two scenarios that were integrated into the mechanical growth model (*Figure 1*) and are compatible with recent experimental observations (*Narasimhan et al., 2021*).

In the first scenario, cells would sense auxin flux through the membrane (also called 'with-the-flux model')(*Feugier and Iwasa, 2006*; *Mitchison, 1997*; *Rolland-Lagan and Prusinkiewicz, 2005*; *Stoma et al., 2008*) and adjust PIN allocation to the plasma membrane in a positive feedback-dependent manner (*Figure 2—figure supplement 1A, B*). Despite that, the exact molecular mechanism behind auxin flux sensing is to be demonstrated and it may involve membrane-associated protein kinases (*Hajný et al., 2020*; *Marhava et al., 2018*; *Michniewicz et al., 2007*). Therefore, we explored a

second scenario for PIN polarization that we named 'regulator-polarizer' (*Figure 2—figure supplement 1C, D*). The regulator-polarizer model implements a potential mechanism behind auxin flux sensing (*Feugier and Iwasa, 2006*; *Mitchison, 1997*; *Rolland-Lagan and Prusinkiewicz, 2005*; *Stoma et al., 2008*). Briefly, a putative regulator (i.e. a general phosphatase [*Michniewicz et al., 2007*]) detects auxin passing through a plasma membrane, it becomes activated and freely diffuses in the plasma membrane. This regulator inhibits a polarizer (e.g. a dedicated protein kinase that phosphorylates PIN *Hajný et al., 2020*; *Michniewicz et al., 2007*) that in turn activates PINs. Therefore, at the side where the concentration of regulator is high enough to overcome the polarizer, no PINs are recruited.

Our model combines exo- or endocytosis and lateral diffusion of PIN proteins into one general trafficking term, which is a crude simplification required to reduce model complexities (see Materials and methods for more details, Auxin transport module description). However, to incorporate quantitative data in the model, PIN recruitment parameters were fitted to the experimentally derived kinetics of PIN trafficking after cell division (*Figure 2K*; *Glanc et al., 2018*). Currently, we do not distinguish in our model between different PIN families (*Sauer and Kleine-Vehn, 2019*), instead, all PINs are distributed according to one of the two PIN polarization scenarios (*Figure 2—figure supplement 1*). The only exception to this general rule is that PINs in the columella are distributed uniformly among membrane sections, to reproduce the observed PIN3 distribution (*Friml et al., 2002*). Given that maximal PINs abundance threshold is the same for all cell types, the fact that columella cells redistribute PINs over the totality of the membrane and not to a specific polar section causes lower overall PINs levels when compared to experimental observations (*Blilou et al., 2005*). Other assumptions of our model are the uniform distribution of AUX/LAX carriers (*Swarup et al., 2001*) in all cell types, and the omission of other transporters such as ABCB transporters (*Cho and Cho, 2013*).

Previous modeling attempts combined auxin transport with tissue mechanics to explain a unidirectional PIN polarity pattern associated with the shoot apical meristem but operated on static non-growing templates (*Heisler et al., 2010*). However, such models have never been applied to root development, in particular in an organ growth context. We combined the biomechanical model (*Figure 1*) and the polar auxin transport component into a coherent mechanistic framework (*Figure 2—figure supplement 2*, *Figure 2—figure supplement 3*), and tested whether this framework is capable of generating the complex PIN polarity network and sustained anisotropic root growth.

Computer model simulations track the growth of the basal part of the embryo (immature root) connected to the aerial segment of the plant embryo (*Figure 2A*). Auxin is introduced into the vascular tissues and exits through the epidermis (*Figure 2A*, *Figure 2—figure supplement 2B*), allowing auxin recycling between the emerging root and the rest of the embryo. This assumption is necessary to recreate a continuous flow of auxin inside the root as observed experimentally (*Möller and Weijers, 2009*). The amount of auxin produced by the source cells does not increase during the simulation; therefore the smaller initial roots contain higher auxin concentrations compared to longer more mature roots, to account for potential hormone dilatation effects in later developmental stages. As the internal turgor pressure balances the cell walls stiffness, auxin at low-to-intermediate concentrations can disrupt this balance by reducing the stiffness of the cell walls, thereby promoting cell wall elongation (*Majda and Robert, 2018*; *Figure 2B*, *Figure 2—figure supplement 2C*). The cell growth rate is regulated by homogenous intracellular auxin concentrations by relaxing the stiffness of the entire cell wall. Auxin however does not directly affect cell growth anisotropy, which is instead determined by growth mechanics (*Figure 1*).

Cell division follows a simple but effective rule: each cell possesses a maximum area attribute so that when a cell reaches a certain threshold it divides into two daughter cells (*Figure 2—figure supplement 2D*). The maximum area is specific for each cell type, so that cell size is maintained consistently for each cell lineage. The orientation of cell division is by default anticlinal and occurs along a division vector passing through the cell centroid and parallel to the AF (*Figure 2—figure supplement 2D*). The time scales of the Quiescent Center (QC) and columella cell growth are very long and these cells divide infrequently (*Kumpf and Nowack, 2015*). To simplify assumptions in our model, neither QC nor columella cells grow or divide.

Time-lapse model simulations predicted the anisotropic auxin-dependent root growth (*Figure 2B and D*) with a growth rate peak located in the apical root meristem (*Figure 2H*) in close agreement with experimental observations(see Figure 2 in *Bassel et al., 2014*). Furthermore, growth anisotropy

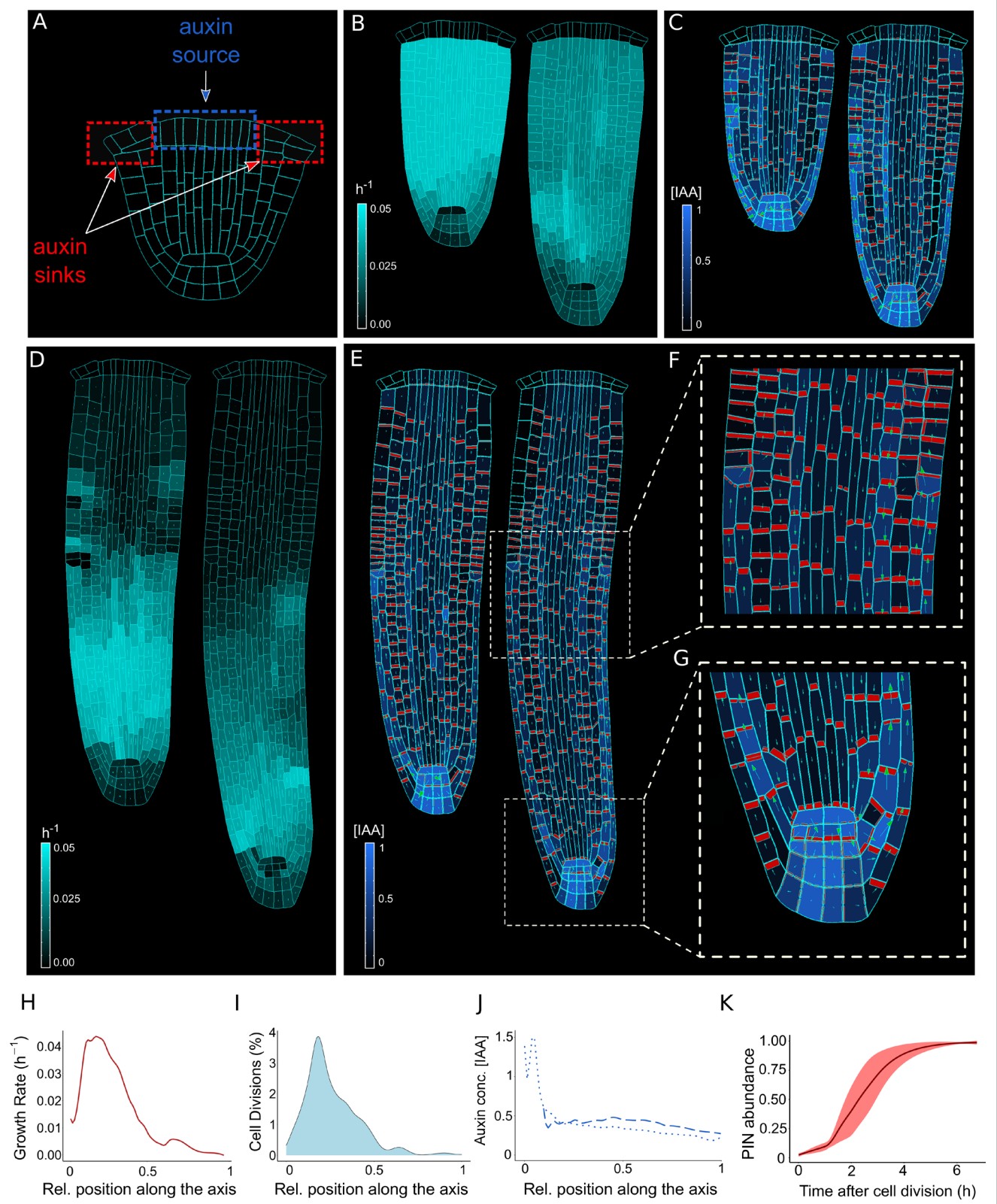

**Figure 2.** The model reproduces realistic root meristem geometry, auxin distribution, and PINs polar localizations using auxin flux scenario. (**A**) Initial embryonic set point. Locations of auxin influx (auxin source, blue) and evacuation (auxin sinks, red) from the embryo are shown. (**B, D**) Model simulations predict a time evolution of cell growth rates (bright cyan color) and principal growth directions (white lines). Ongoing cell division events are marked by black regions. (**C, E**) Dynamics of auxin distribution (blue color), auxin flow direction (arrows,) and PIN localizations (red). (**F, G**) Zoom on basal meristem

*Figure 2 continued on next page*

*Figure 2 continued*

(**F**) and root apical meristem (**G**). The model correctly reproduces very detailed PINs localizations including bipolar PIN2 localization in the cortex (**F**). (**H–J**) Profiles of average values of interest across all cell files along the longitudinal axis. (**H**) Growth rate profile along the root axis. The fastest-growing region is located in the apical meristem as observed experimentally (*Bassel et al., 2014*). (**I**) Cell division frequencies along the root axis. The majority of cell divisions occur in the apical meristem. (**J**) Auxin concentration in the vascular tissues (dashed blue line) and auxin concentration in the non-vascular tissues (external tissues and the root tip, dotted blue line) along the root axis. Most of the auxin is concentrated in the root tip as observed in experiments (*Overvoorde et al., 2010*). Time-lapse profile of PINs re-localization on the membranes after cell division event. PINs re-localization is completed in approximately 5–6 hr after cell division (*Glanc et al., 2018*). All simulations have been run until 1500 time steps were reached.

The online version of this article includes the following video and figure supplement(s) for figure 2:

**Source data 1.** Source data used to generate *Figure 2H-K*.

**Figure supplement 1.** Comparison between 'auxin-flux' and 'regulator-polarizer' models for PIN polarization.

**Figure supplement 2.** Model schematics.

**Figure supplement 3.** Schematic diagram of the root model.

**Figure supplement 4.** Anisotropy index measured along the proximo-distal axis.

**Figure supplement 4—source data 1.** Source data used to generate *Figure 2—figure supplement 4*.

**Figure supplement 5.** The model can reproduce realistic root meristem geometry, auxin distribution, and PINs localization using the 'regulator-polarizer' model scheme.

**Figure supplement 5—source data 1.** Source data used to generate *Figure 2—figure supplement 5H-K*.

**Figure supplement 6.** Cell division rule testing.

**Figure supplement 7.** Comparison between the reference model with an alternative model simulation in which the contribution of the anisotropy factor (AF) to PIN localization is omitted at the start the of simulation, using the 'auxin-flux' (**A–B**) and the 'regulator-polarizer' (**C–D**) models, respectively.

**Figure 2—video 1.** Model simulations obtained with the auxin-flux model, related to Figure 2.
https://elifesciences.org/articles/72132/figures#fig2video1

**Figure 2—video 2.** Model simulations obtained with the regulator-polarizer model, related to Figure 2—figure supplement 5.
https://elifesciences.org/articles/72132/figures#fig2video2

(*Figure 2D* and *Figure 2—figure supplement 4*) and associated cell division patterns (*Figure 2I*) correlate with the predicted auxin distributions (*Figure 2C, E and J*), producing auxin-guided aniso-tropic growth and polar pattern of PIN localization (*Figure 2G*, *Figure 2—figure supplement 5G*, *Figure 2—videos 1 and 2*), in both PIN polarization scenarios (*Figure 2—figure supplement 2E*). The predicted auxin maximum forms close to the QC (*Figure 2G and J*) and represents the equilibrium between auxin reaching the root tip from the vascular tissues and auxin leaving the root tip to the outermost tissues. Previous evidence showed that this position of auxin maximum is necessary for the correct organization of the meristem (*Petersson et al., 2009*).

Nevertheless, to maintain a correct shape of the root tip additional assumptions were necessary to regulate the cells belonging to the stem cell niche, which are known to follow alternative division rules (*Fisher and Sozzani, 2016*), cortex/endodermis initial daughter(CEID) cells and the epidermis/lateral root cap initials divide periclinal and alternatively anticlinal/periclinal, as previously described (*Choi and Lim, 2016*). We tested the importance of this experimentally-supported assumption by demonstrating that in its absence the model produced an incorrect pattern of cell divisions in ground tissues and altered root morphology (*Figure 2—figure supplement 6*).

Our combined mechano-biochemical model was able to reproduce a complex PIN polarization network from an initially non-polar scenario (*Figure 2E–G*, *Figure 2—figure supplement 5E-G* and *Figure 2—videos 1 and 2*). This dynamic network includes rootward PIN localization in vascular tissues and shootward localization in the outermost epidermis that closely follow experimentally observed patterns (see Figure 1 in *Blilou et al., 2005* and Figure 2 in *Tanaka et al., 2006*).

The model predicts that PIN polarity patterns emerge from mechanical constraints, auxin flow, and auxin-mediated growth – likely the elements of the same feedback mechanism. To further illustrate this entanglement between mechanics and biochemical components, we tested the importance of AF for PIN trafficking (*Figure 2—figure supplement 7*). We simulated an alternative model version in which AF was completely removed from the factors regulating PIN trafficking (since the beginning of the simulation). For both the 'auxin flux (*Figure 2—figure supplement 7A, B*) and the 'regulator-polarizer' (*Figure 2—figure supplement 7C, D*) scenarios, the absence of mechanical constraints

regulating PIN localization results in incorrect auxin/PIN distribution, with the disappearance of the auxin maximum and a general lack of PIN polarity. This important finding suggests a strong involvement of mechanical deformation in the root cell polarity patterning mechanisms.

Another intriguing emergent property of the model was the bidirectional (shootward and rootward) 'bipolar' localization of PIN proteins in the cortex tissues (*Figure 2F* and *Figure 2—figure supplement 5F*) in the transition region that is marked by the termination of lateral root cap (LRC). This 'bipolar' PIN localization in the cortex has been previously observed in experiments close to the transition zone (*Ötvös et al., 2021*; *Sauer et al., 2006*). Yet, the function of this phenomenon remains largely unknown. Model simulations suggest that the bipolar cortex PIN localization is likely the result of the conflict between the shootward auxin flow from LRC/epidermis and the rootward auxin flow in the vascular tissues (*Figure 2F* and *Figure 2—figure supplement 5F*). However, we observed a subtle difference between the 'auxin-flux' (*Figure 2F*) and the 'regulator-polarizer' (*Figure 2—figure supplement 5F*) scenarios regarding the PIN lateralization pattern. The likely explanation for these small differences is that the 'auxin-flux' scenario allocates PINs based on global flux patterns whereas the 'regulator-polarizer' scenario depends on local auxin concentrations at a given membrane segment.

Taken together, computer simulations indicate a plausible mechano-biochemical model that accounts for auxin-dependent anisotropic root growth and PIN polarity establishment.

## Shoot-independent root growth requires auxin reflux, local auxin production, and balance in auxin levels

Our model simulations reconstitute the complex PIN polarity network in the simulated root growth (*Figure 2F–G*, *Figure 2—figure supplement 5F-G*), suggesting the presence of lateral auxin transport from the external tissues (epidermis and LRC) into the cortex and the stele (*Figure 2E*, *Figure 2—figure supplement 5E*). This 'bipolar' PIN localization (*Figure 2F*, *Figure 2—figure supplement 5F*) could drive polar auxin redistribution towards inner tissues, that is consistent with the phenomenon described as the reflux loop (*Benková et al., 2003*; *Grieneisen et al., 2007*; *Paponov et al., 2005*). Although not covered by our model, this lateral auxin transport between the epidermis and cortex might be further enhanced by plasmodesmata-dependent diffusion (*Mellor et al., 2020*). Yet, it is a directionality of transport mediated by PINs that is critical for the growth coordination of adjacent epidermis and cortex tissues (*Ötvös et al., 2021*). How this reflux phenomenon would operate on realistic tissue geometries constrained by growth mechanics remains, however, unclear.

To further investigate the importance of a dynamic PIN localization network for the sustained growth of the root, we performed model simulations by artificially preventing lateral auxin transport (*Figure 3A and B*). We found that a negligible amount of auxin enters the cortex, but no lateral auxin influx originated from the epidermis. Additionally, the bipolar PIN localization was absent in these 'no-reflux' simulations (*Figure 3C* and *Figure 3—video 1*) compared to the reference model (*Figure 3D* and *Figure 3—video 1*). This finding confirms the importance of PIN-mediated lateral transport for auxin redistribution in inner root tissues. However, the lack of auxin recycling in the meristem does not seem to significantly reduce root growth rates as long as auxin is supplied from the shoot (*Figure 3E*). Therefore, to investigate the role of shoot-derived auxin source in the root growth, we artificially separated the root from the rest of the plant by removing the shoot-derived auxin source (*Figure 3A–B*, bottom panel). In this simulation where there was neither reflux nor bipolar PIN localization, root growth could not be sustained over a prolonged time and the auxin inside the root eventually disappeared (*Figure 3E*). On contrary, the reflux scenario allows for the maintenance of auxin levels over a prolonged period even without the shoot-derived auxin source being removed. Root growth can be further strengthened by incorporating auxin biosynthesis in the QC cells (*Stepanova et al., 2008*), which in theory could sustain root growth almost indefinitely (*Figure 3E–F*).

These results together indicate that the presence of an auxin reflux loop mediated by bidirectional PIN transport and diffusion in the cortex/epidermis is capable to sustain root growth for prolonged periods.

Keeping the correct balance in auxin levels might also be important to sustain root growth mechanics. To test how alterations in auxin levels alone would impact root growth dynamics, we successively simulated a series of external auxin applications for 6 hr by increasing the overall auxin content of the root (*Figure 4A–B* and *Figure 4—video 1*). Model simulations show the sequential inhibition and reinstatement of root growth after cyclic auxin removal (*Figure 4C*). A similar trend was

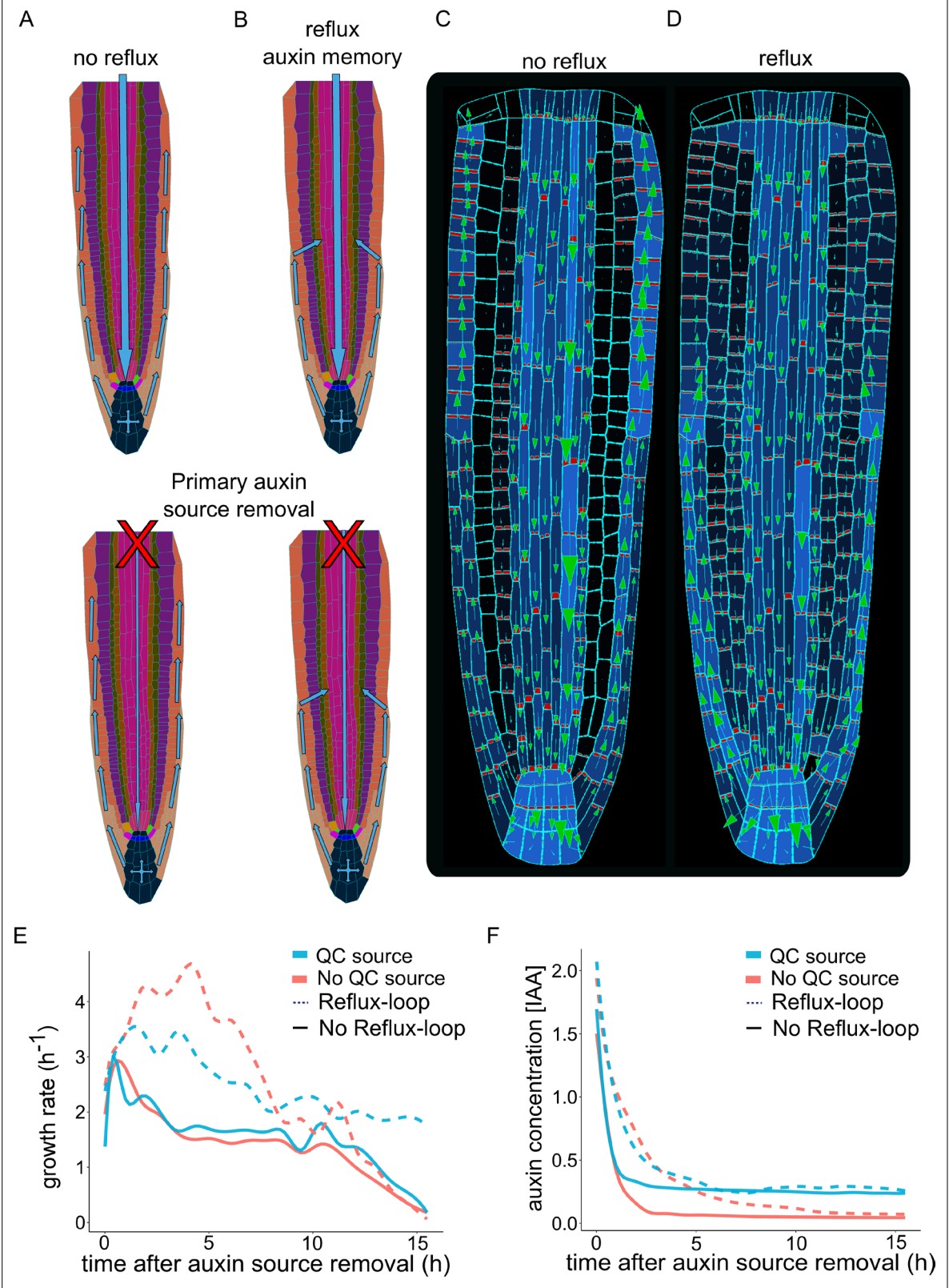

**Figure 3.** Independent root growth requires auxin reflux and local auxin production. (**A–D**) Schematics (**A and B**) and model simulations (**C and D**) with the disabled auxin reflux-loop (**A, C**) or wild-type-like scenario with self-emerging reflux (**B, D**). Only the in reflux scenario auxin moves from the epidermis back into the vascular tissues sustaining the long-term root growth. (**E**) Growth rate profiles of model simulations after primary auxin source removal, in four different scenarios. The plot shows the total root growth rate over time. In the absence of an auxin reflux-loop, the root is unable

*Figure 3 continued on next page*

*Figure 3 continued*

to sustain growth for a long period (solid lines) even if a secondary auxin source in the root tip was introduced (solid blue line). On the contrary, the presence of an auxin reflux-loop sustains the root growth for prolonged periods (dotted lines), further augmented by the presence of a secondary auxin source in the root tip (dotted blue line). (**F**) Auxin concentration profiles of model simulations after primary auxin source removal. The plots show the average radial auxin concentration among the root cells. In the absence of an auxin reflux-loop, the average auxin concentration in the root quickly drops to zero (solid red line). Alternatively, the presence of an auxin reflux-loop allows the root to maintain an auxin reserve for prolonged periods (dotted blue line). The presence of a secondary auxin source in the root tip preserves an auxin reservoir and sustains root growth in the long term (blue lines). The model simulations have been run for 1000 time steps.

The online version of this article includes the following video and figure supplement(s) for figure 3:

**Source data 1.** Source data used to generate *Figure 3E and F*.

**Figure 3—video 1.** Model simulations with the enabled/disabled auxin reflux-loop and with/without local auxin production in the QC, related to Figure 3.

https://elifesciences.org/articles/72132/figures#fig3video1

observed for a shorter period of stimulation (*Figure 4—figure supplement 1*). Notably, these model predictions replicate the experimentally observed temporal inhibition of root growth by external auxin applications (see Figure 1f in *Fendrych et al., 2018*).

Our analysis indicates that our root model can correctly recapitulate experimentally observed modulation of root growth response to externally applied auxin. Also, our model suggests that the balance in auxin content maintained by the network of PIN polarity is critical for the sustained growth of the root meristem.

## Model simulations reproduce experimentally observed phenotypes of auxin-mediated growth and mechanical perturbations

Our analysis indicates that the mechano-biochemical framework for root meristem growth could be potentially used to test dynamic perturbations of root growth, such as genetic alterations and mechanical manipulations, guiding the further design of wet-lab experiments. To test the predictive power of our model we investigated how alterations of auxin transport parameters could perturb patterning dynamics and whether these predictions would match experimental observations.

PIN2 is an important auxin efflux carrier expressed in the most external root tissues: cortex, epidermis, and lateral root cap (*Adamowski and Friml, 2015*), and steers root gravitropic responses (*Rahman et al., 2010*). PIN2 loss-of-function results in defective gravitropic response largely because of disrupted auxin transport dynamics (*Dhonukshe et al., 2010*). To test whether our model could predict the alterations of auxin distribution observed in *pin2* mutants, we performed computer simulations by reducing PIN expression rate in the epidermis, cortex, and lateral root cap (*Figure 5A–B* and *Figure 5—video 1*). The reduced levels of PINs in these outermost tissues resulted in auxin accumulation in the lateral root cap on both sides of the root (*Figure 5B*), which was absent in the wild-type simulations (*Figure 5A*). These predictions mimic experimental observations of *pin2* knockdown mutant (see Figure 2f in *Liu et al., 2018*). Similarly, the reduced expression of PIN-dependent transport in the inner vascular tissues in our model predicts the alteration of auxin distribution and growth defects (*Figure 5—figure supplement 1A* and *Figure 5—video 2*). This prediction could reflect the scenario of reduced levels of vascular PINs (PIN knockdown) as opposed to the full knockout which is lethal (*Vieten et al., 2005*). Finally, we tested how a general knockdown of auxin cellular influx would impact root growth. Severely reducing auxin cellular influx (by 90 % reduction of AUX/LAX expression) led to lower auxin content, reduced sensitivity to auxin, and thereby slow root growth (*Figure 5—figure supplement 1B* and *Figure 5—video 3*) as previously suggested (*Inoue et al., 2016*; *Liu et al., 2018*).

Next, we tested how local mechanical disruptions of QC, root tip, and LRC would alter the model outcome, and whether this outcome agrees with experimental observations. The QC is a small group of cells (four to seven in the *A. thaliana* root) located in the middle of the root apical meristem (*Doerner, 1998*). The QC divides infrequently and grows at an extremely low rate (*Nawy et al., 2005*). The QC is known to be the source of signals that inhibits differentiation of the surrounding stem cells (*van den Berg et al., 1997*). QC cells define the correct location of the stem cell niche but also behave as independent cells by self-renewing and replenishing initials that have been displaced from their position (*Kidner et al., 2000*). QC laser ablation is not lethal for the root as a new QC and stem niche

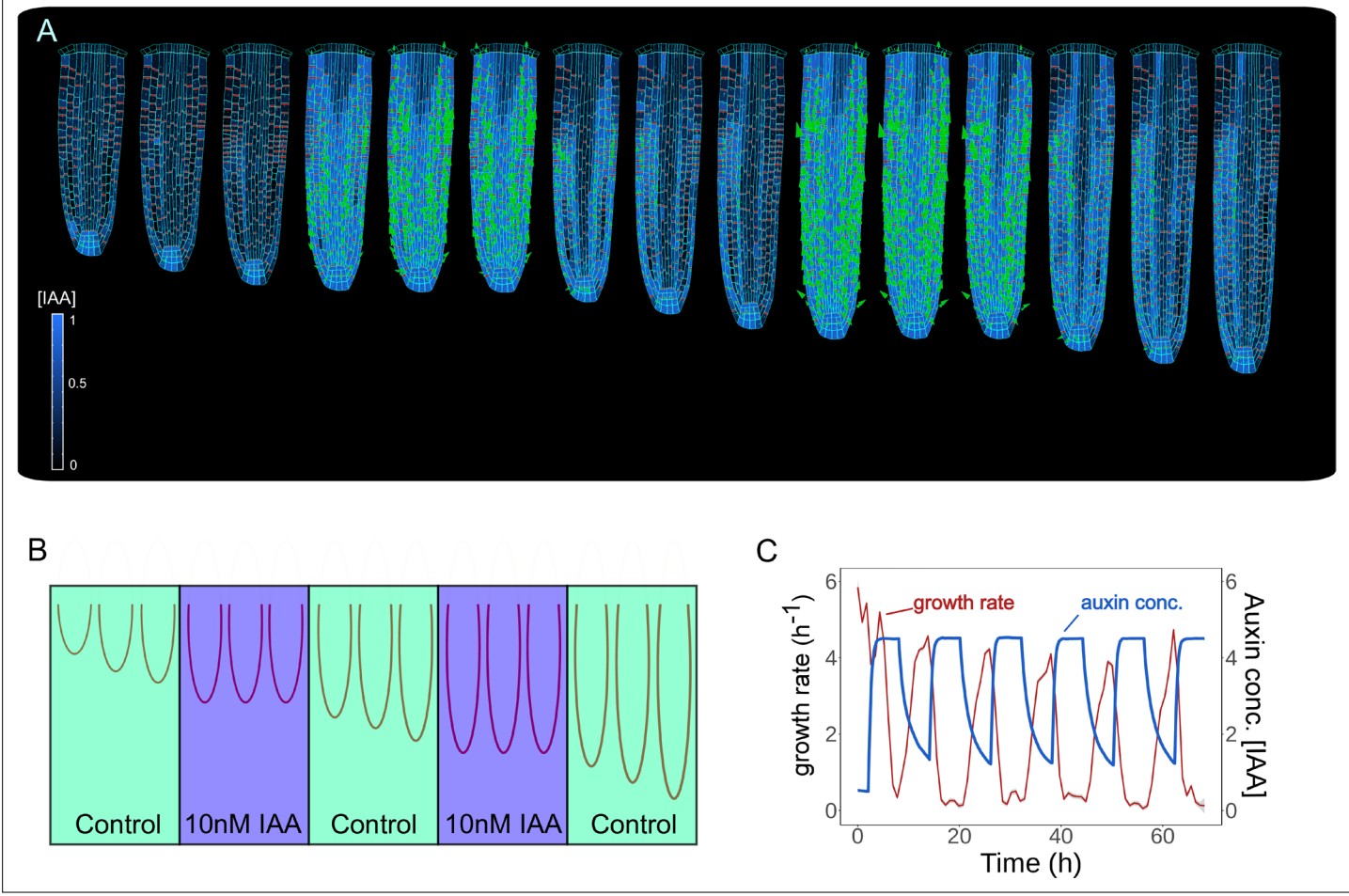

**Figure 4.** Model predictions reproduce reversible inhibition of root growth by externally applied auxin. (**A**) Successive application of external auxin in model simulations according to a predefined cycle. Root growth is inhibited by the introduction of high amounts of auxin and subsequently restored after the external application is stopped as seen in experiments (*Fendrych et al., 2018*). (**B**) Schematic of the in silico experiment. To simulate auxin treatment as described in *Fendrych et al., 2018*, we introduced external auxin inside the root (by inducing excessive auxin synthesis at individual cell level) at predefined time points to inhibit root growth and subsequently removed to allow root growth re-establishment. (**C**) Time-lapse profile of root growth rate (red line) and average cell auxin concentrations (blue line). The cycles of external auxin applications inhibit and restore root growth, respectively. The simulation has been run for 1500 time steps.

The online version of this article includes the following video and figure supplement(s) for figure 4:

**Source data 1.** Source data used to generate *Figure 4C*.

**Figure supplement 1.** The reversible inhibition of root growth by external auxin applications.

**Figure supplement 1—source data 1.** Source data used to generate *Figure 4—figure supplement 1*.

**Figure 4—video 1.** Successive application of external auxin in model simulations according to a predefined cycle, related to Figure 4.
https://elifesciences.org/articles/72132/figures#fig4video1

are quickly reestablished a few cells above the wound in correlation with increased auxin accumulation (*Sabatini et al., 2003*). We replicated the same experiment in silico by removing the two QC cells from the model during a simulation (*Figure 5—figure supplement 1C* and *Figure 5—video 4*). Compared to the wild-type simulations (*Figure 5A*), the typical auxin accumulation in the root tip is depleted, and auxin reflux in the LRC was significantly reduced, while most of the auxin coming from the shoot tends to concentrate in the cells above the ablation, exactly as observed in experiments (see Figure 5 in *Reddy et al., 2007*). Similarly, removal of the LRC led to sharp auxin accumulation in the root tip (*Figure 5C* and *Figure 5—video 5*), largely matching empirical data (*Tsugeki and Fedoroff, 1999*).

 *A. thaliana* roots can survive not only after QC ablation but even after the complete excision of the root tip, as the plant can regenerate a new root tip including a complete new root apical meristem

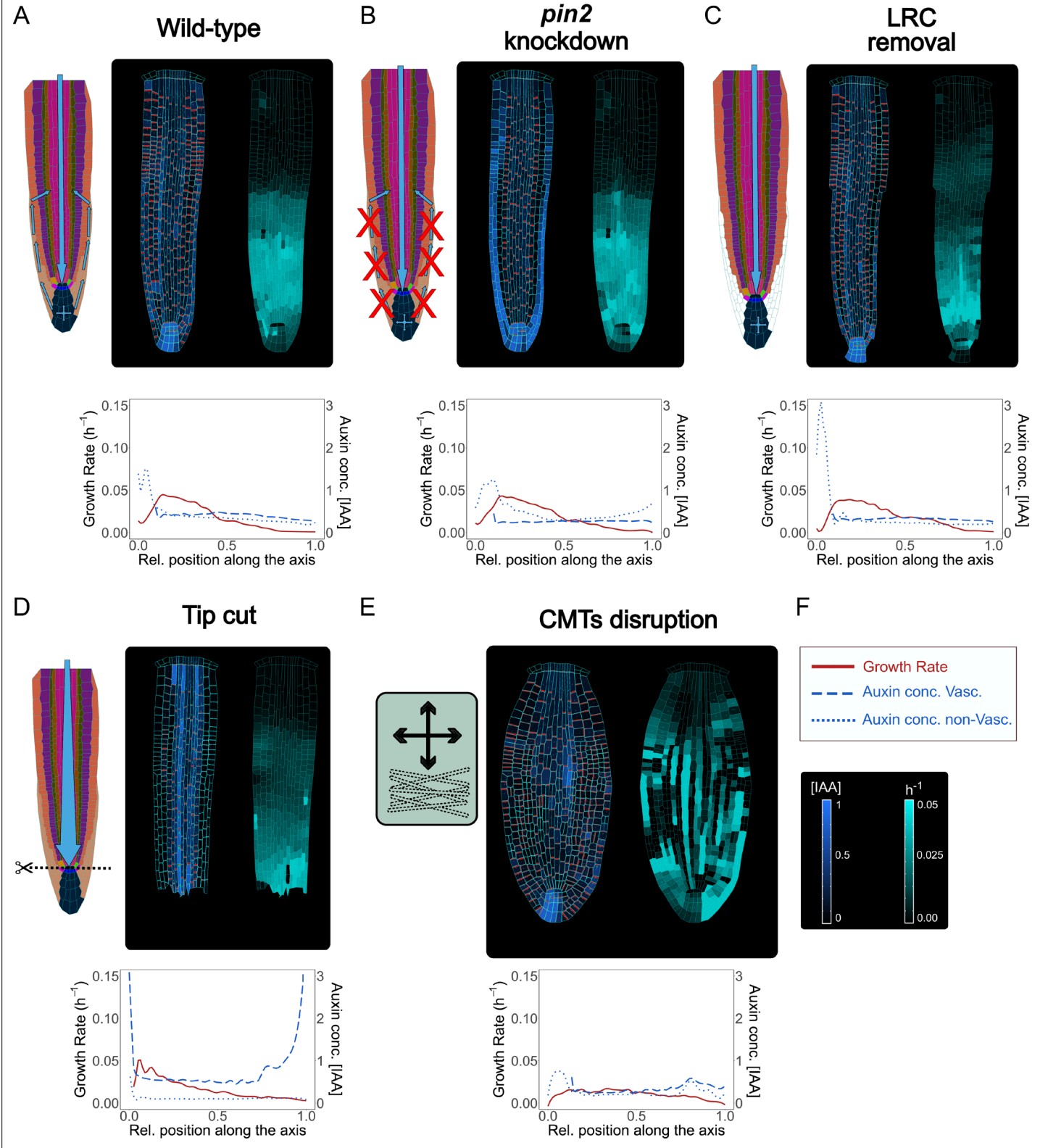

**Figure 5.** Model simulations recapitulate experimentally observed phenotypes through genetic, pharmacological, and mechanical perturbations. (**A**) Reference model simulation of the wild-type scenario. The figure displays a schematic representation of the auxin flow inside the root (left picture), cell growth rate (right picture). The bottom graph shows the profiles of auxin concentration in the vascular tissues (dashed blue line), auxin concentration in the non-vascular tissues (dotted blue line,) and growth rate (red line) along the root axis. (**B**) Model simulation of the *pin2* knockdown mutant. In silico

*Figure 5 continued on next page*

*Figure 5 continued*

pin2 mutant shows strongly reduced PINs expression in the lateral root cap, epiderm, is, and cortex. Note that acropetal auxin flow is severely affected and auxin tends to accumulate in the lateral tissues as observed in experiments (*Dhonukshe et al., 2010*). (**C**) Mechanical removal of lateral root cap resulted in the strong accumulation of auxin inside the root tip, largely because auxin cannot flow anymore shootward through outermost tissues whereas growth rate was not significantly affected. (**D**) Simulation of root tip cutting. Removing the root tip results in a general increase of auxin level in the central vascular tissues, as a consequence of the disappearance of acropetal auxin flow. PINs localization in the external tissues is also affected due to the loss of incoming auxin flow. (**E**) Simulated CMTs disruption (i.e. oryzalin treatment or similar) on root growth and polarity. CMTs disruption was simulated by inducing a fast degradation of the anisotropy factor. Cells lose polarity and growth anisotropy, causing the root to expand and bulge radially as observed in experiments (*Baskin et al., 1994*). Notice that the top cell row is considered to be a static attachment of the root to the substrate and therefore its growth is not affected during the simulation. (**F**) Legend and scale bars of auxin concentration and cell growth rate. 'Auxin conc. Vasc.' indicates auxin concentration in the vascular central tissues (the vascular cells and the pericycle), while 'Auxin conc. non-Vasc.' indicates auxin concentration in the remaining external tissues and the root tip. The simulations have been run for 1500 time steps.

The online version of this article includes the following video and figure supplement(s) for figure 5:

**Source data 1.** Source data used to generate *Figure 5A-E*.

**Figure supplement 1.** Model simulations recapitulate some of the experimentally observed phenotypes, related to *Figure 5*.

**Figure supplement 1—source data 1.** Source data used to generate *Figure 5—figure supplement 1A–1C*.

**Figure supplement 2.** Model simulations using alternative wild-type embryo templates.

**Figure supplement 3.** Model parameters sensitivity.

**Figure 5—video 1.** Model simulations of the *pin2* knockdown mutant, related to Figure 5B.
https://elifesciences.org/articles/72132/figures#fig5video1

**Figure 5—video 2.** Model simulations of the vascular PINs reduction, related to Figure 5—figure supplement 1A.
https://elifesciences.org/articles/72132/figures#fig5video2

**Figure 5—video 3.** Model simulations of the *aux1* knockdown mutant, related to Figure 5—figure supplement 1B.
https://elifesciences.org/articles/72132/figures#fig5video3

**Figure 5—video 4.** Model simulation of QC ablation, related to Figure 5—figure supplement 1C.
https://elifesciences.org/articles/72132/figures#fig5video4

**Figure 5—video 5.** Model simulation of lateral root cap ablation, related to Figure 5C.
https://elifesciences.org/articles/72132/figures#fig5video5

**Figure 5—video 6.** Model simulations of root tip cutting, related to Figure 5D.
https://elifesciences.org/articles/72132/figures#fig5video6

**Figure 5—video 7.** Model simulation of mechanics disruption on root growth and polarity, related to Figure 5E.
https://elifesciences.org/articles/72132/figures#fig5video7

---

(*Efroni et al., 2016*). Since the stem cell niche is lost with excision, the regeneration process relies on other pluripotent dormant cell types available in the remaining stump (*Sugimoto et al., 2010*). We tested if we could replicate this experiment by removing the entirety of the root tip during the simulation (*Figure 5D* and *Figure 5—video 6*). Compared to QC ablation (*Figure 5—figure supplement 1C* and *Figure 5—video 4*), the removal of the root tip displays an even more radical effect on auxin patterning dynamics (*Figure 5D* and *Figure 5—video 6*). Auxin signal was strongly increased in the vascular tissues and auxin reflux in the lateral tissues was absent; again, model predictions closely match experimentally observed patterns (see Figure 1 in *Matosevich et al., 2020*).

Additionally, we explored whether simulated chemical perturbation of core mechanics would reproduce the experimentally observed root phenotypes. CMTs organization can be modulated by chemical treatments which cause microtubules depolymerization and stimulate the radial expansion of roots (*Baskin et al., 1994*). We simulated CMTs disruption by implementing a gradual reduction of the AF during root growth (*Figure 5E* and *Figure 5—video 7*). The simulated root displays a marked radial swelling, more evident in the center of the meristem and much less pronounced in the root tip (*Figure 5E*). Several cells divide irregularly, and the organ loses its anisotropic shape. As a consequence of this, PINs polarities become more irregular, and asymmetric auxin distribution is eventually lost (*Figure 5E*). Also, we tested the model robustness concerning cellular geometry and key model parameters that control PIN polarity and auxin effect on cell growth rates. Choosing alternative staring geometries (*Nieuwland et al., 2016*; *Scheres et al., 1994*) has no visible impact on root anisotropy, auxin distribution, and PIN patterns in the simulations (*Figure 5—figure supplement 2A,*

*B*). Similarly, we found our model predictions to be generally robust for a plausible range of parameter values (*Figure 5—figure supplement 3A, B*).

These results demonstrate that our model can reproduce various root meristem phenotypes including several auxin transport mutants, and mechanical or chemical manipulations of root tissue geometry. Thus, our model could provide a useful tool for guiding wet-lab experimental designs.

## Discussion

Computer models have become a powerful tool for wet-lab scientists to quickly explore possible mechanisms and theories underlying organ growth patterns and thus to guide and design effective experimental strategies. To date, computer models of root development have been instrumental in understanding root maturation and zonation through biochemical processes integrated over non-growing (*Band et al., 2014*; *Rutten and Ten Tusscher, 2019*) or idealized templates (*Grieneisen et al., 2007*). However, little to no attempts were made to couple mechanisms of cell polarity establishment and realistic tissue biomechanics at single-cell resolution to mechanistically understand how root growth and cell polarities are established from small populations of differentiated cells.

Here, we took advantage of an efficient modeling technique called Position-Based Dynamics (*Müller et al., 2007*) to resolve biomechanics of root growth at single-cell resolution, while simultaneously incorporating biochemical reactions that guide auxin production, distribution, and polar transport across tissues. Our mechanistic cell-based model successfully reproduces important elements of the root meristem morphology including cell polarity organization, auxin distribution, and sustained anisotropic root growth. In this framework, root growth patterns result from local cell growth activities and direct cell-to-cell communication mediated by auxin without the need for global regulators or polarizers. Furthermore, our model demonstrates that auxin influx from the LRC and subsequent 'bipolar' PINs localization in the cortical tissues may be important elements for sustained auxin-dependent root growth.

In particular, we found that PIN polarity depends on the auxin flow entering the cell but also on mechanical constraints, and a plausible molecular mechanism for PIN polarization based on a putative kinase/phosphatase regulation was proposed (*Hajný et al., 2020*; *Michniewicz et al., 2007*; *Weller et al., 2017*). We further show that our model can be extended to address many aspects of root development and organogenesis including root cells ablation, root response after chemical treatment, and genetic mutations. As the quantitative model predictions largely reproduce experimental observations, our model could be a useful tool to predict the phenotypes of various mutants and test the effects of perturbations such as chemical treatments, gene knockdown, or mechanical alterations, guiding further the effective design of wet-lab experiments. In the future, our model could be expanded to address additional mechanisms of root zonation (*Ivanov and Dubrovsky, 2013*), stem cells differentiation (*Sabatini et al., 2003*), lateral root initiation (*Perianez-Rodriguez et al., 2021*), and auxin flux through plasmodesmata (*Mellor et al., 2020*). These results support the robustness of the model and allow the possibility for modular extensions of the current framework to account for further complexities; for instance, the action of other hormones and postembryonic regulatory mechanisms like gravitropism and phototropism. Furthermore, this type of model framework can be employed to model other plant organs at cellular resolution as both auxin and mechanics are important general aspects of organogenesis in plants.

Nevertheless, the current framework relies on several simplifications and assumptions; we specified ad-hoc rules for cell division in the stem cell niche patterning, we simplified the combined action of cytoskeleton components such CMTs, actin, and cell wall composition, and chose an initial root tip organization. Future improvements of the current model should focus on the regulation of cell differentiation, auxin-control of stem cell niche maintenance, detailed protein trafficking, tissue-specific expression of auxin transporters, root zonation, and tropism by integrating new experimental insights. An important aspect missing in the current model is rapid cell elongation; this would require the implementation of dynamic tissue remeshing and the preservation of mechano-chemical information.

Taken together, our study highlights the general design principles underlying root growth organization determined by local interactions between directional transport of auxin, auxin-dependent cell elongation, cell polarization, and biomechanical stimuli, and presents a step forward toward quantitative subcellular models of plant organogenesis which could serve as a next-stage platform to develop novel traits of high socio-economic importance.

## Materials and methods
### Cellular mesh segmentation and processing
The process of segmentation of microscopy images with MorphoGraphX is broken into several steps:

- The microscope images of an *A. thaliana* embryo without the aerial parts contain black background and color cell borders with high contrast
- Images are loaded as the MorphoGraphX Image Stack structures.
- "The Mesh-Creation-Mesh Cutting Surface" process is executed inside the MorphoGraphX framework to create an initial mesh of the root.
- The initial mesh is subdivided several times to increase the detail level.
- Cell borders are projected over the mesh to mark individual cells.
- Stack of images is then segmented using the standard MorphoGraphX pipeline (*Barbier de Reuille et al., 2015*).
- Mesh was converted into cells with Tools-Cell Maker-Mesh 2D-Tools-Polygonize Triangles. "Max Length" parameter was set to zero.
- A final meshed model was smoothed for irregularities and artifacts and scaled appropriately.

### General model description
The root model was created using MorphoDynamX, the second generation of the MorphoGraphX software (*Barbier de Reuille et al., 2015*). This modeling framework is based on an advanced data structure called Cell Complexes (*Karwowski and Prusinkiewicz, 2004*; *Prusinkiewicz and Lane, 2013*) that expands the previous methodology called Vertex-Vertex complexes (*Federl and Prusinkiewicz, 1999*) to model subdividing geometries in two and three dimensions. MorphoDynamX provides the user interface and API interface to the Cell Complexes. Cells are represented as triangulated polygons obtained through the segmentation and mesh processing pipeline described in(Cellular mesh segmentation and processing). Cells are composed of vertices, edges, and faces. Each of these three base elements (vertices [0 dimension], edges [one dimension], and faces [two dimensions]) has its biological interpretation and possesses different attributes and properties that allow the model to run and produce dynamically growing organ structures. Perimeter edges represent the cell membrane while internal edges mimic the cell cytoskeleton (i.e. actin, CMTs). These edges store both mechanical and biochemical attributes.

To create a continuous flow and recycling of auxin inside the root we assumed that the cells at the very top of the mesh are considered either sources or sinks; the central row of cells represent the source coming from the aerial side of the embryo, while the most external epidermal cells act as sinks by moving auxin from the root back to the embryo (*Möller and Weijers, 2009*). The mechanics of root growth are implemented using Position-Based Dynamics (PBD) (*Müller et al., 2007*) (see Position-based dynamics implementation). PBD simulates physical phenomena such as material deformation, fluids, fractures, or material rigidness (*Müller et al., 2007*). PBD allows overcoming the typical limitations of force-based models by directly updating positions of vertices based on a set of biologically sound constraints. Whereas chemical processes are numerically solved using the Euler integration method (*Butcher, 2007*).

### Time-lapse confocal imaging of young seedlings
Confocal laser-scanning micrographs of 35 S::PIP2-GFP transgenic lines were obtained as published elsewhere(Zhu, Q. et al, 2019). Briefly, seeds were stratified for 3 days, seed coat was removed and peeled embryos were imaged using a vertical Zeiss LSM700 microscope with a 488 nm argon laser line for excitation of GFP fluorescence. Emissions were detected between 505 and 580 nm with the pinhole at 1 Airy unit, using a 20 x air objective. Images were taken every 20 min and Z-stack maximal projections were done using ImageJ software.

### Computer model assumptions
The root of *A. thaliana* is made of several radially organized layers of morphologically similar cells that can be distinguished in radial and longitudinal sections (*Dolan et al., 1993*; *Scheres et al., 1994*). The central vascular tissue is composed of a bundle of thin and elongated cells surrounded by the pericycle - a cylindrical sheath protecting the stele. The pericycle is also the origin of emerging lateral organs (*Lavenus et al., 2013*; *Péret et al., 2009*). The central cylinder (stele) is enclosed by three

adjacent tissues endodermis, cortex, and epidermis. The gravity-sensing columella is located at the very tip of the root and is composed of four layers of differentiated cells (*Kumpf and Nowack, 2015*). The meristem of the mature root is covered by the lateral root cap which protects the meristem and is periodically shed and replaced by new emerging layers (*Di Mambro et al., 2019*; *Kumar and Iyer-Pascuzzi, 2020*). Finally, the root tip stores a group composed of undifferentiated stem cells that divide asymmetrically and replenish the upper sections of individual tissues (*Stahl and Simon, 2009*). Therefore, this precise spatial-temporal arrangement of tissues in the root requires the coordination of cell polarity, anisotropic growth, and asymmetric cell divisions.

Auxin-driven root growth of *A. thaliana* has been intensively studied in the last years, and it is known to be one of the major players in root development (*Ljung, 2013*). Auxin distributes along the root through a tightly controlled mechanism and its disruption results in organ growth failure (*Truman et al., 2010*). Auxin synthesis and homeostasis are thought to be the other major contributor to cell elongation (*Velasquez et al., 2016*). The main source of auxin during globular root embryogenesis comes from the shoot and tends to accumulate in vascular tissue, root tip, and epidermis (*Robert et al., 2015*; *Smit and Weijers, 2015*). Some aspects of auxin transport by PIN efflux carriers are well understood, but the mechanisms connecting polar transport and auxin effect on root growth remain puzzling (*Adamowski and Friml, 2015*; *Habets and Offringa, 2014*).

Based on known characteristics of *A. thaliana* root, we integrate the following biological assumptions in our models:

- The root is composed of cells categorized into different lineages: QC, Columella Initial, Columella, Epidermal/LRC initial, Cortex/Endodermis Initial (CEI), Cortex/Endodermis Initial daughter (CEID), Lateral Root Cap (LRC), Epidermis, Endodermis, Cortex, Pericycle, and Vascular (*Benfey et al., 2010*; *Nawy et al., 2005*).
- Cell expansion is described according to the acid-growth hypothesis (*Rayle and Cleland, 1992*). Cells are under constant osmotic pressure, and their expansion is prevented by a stiff cell wall with viscoelastic properties. Cells can be considered as incompressible objects. Cell walls possess a strong extensional stiffness at very low or negligible auxin concentration (but also at very high auxin concentration) which prevents cell expansion. Auxin (indole-3-acetic acid, IAA), induces acidification of the cell wall activating a range of enzymatic reactions which modifies the extensibility of plant cell walls, allowing the cell expansion (*Cosgrove, 2000*; *Hager et al., 1991*).
- The mechanical deformation of the cell walls controls the reorientation of the anisotropy factor (AF) and consequently restricts growth along the perpendicular axis to that deformation, creating feedback-dependent reinforcement of anisotropic cell elongation (see Anisotropy factor (AF) and cell polarity). This process can be summarized in the following diagram:

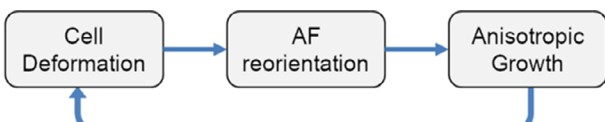

**Scheme 1.** Process diagram.

- It is a simplification in the model and could be replaced in the future with tensile stresses. It has been shown that microtubules are often perpendicular to the maximal cell walls strain and they usually align parallel to predicted maximal tensile stress direction, and the latter is considered to be the best predictor for microtubules reorientation (*Hamant et al., 2019*). Microtubules in turn direct microfibrils deposition which restricts cell expansion in a determined direction producing anisotropic growth (*Bou Daher et al., 2018*).
- Cell division occurs according to cell polarity or cell type specification (*Figure 2—figure supplement 2D*).
- Auxin flows into the root from the aerial section of the embryo through the vascular tissues (*Petrásek and Friml, 2009*). Auxin can also be locally produced in the root apical meristem (*Kerk et al., 2000*; *Stepanova et al., 2008*).
- Auxin diffuses inside cells and all over the intercellular space. Auxin is also actively transported by auxin influx and efflux carriers (*Hosek et al., 2012*). Auxin exchange between cells is not direct but it occurs through the intercellular space which is not visually displayed but

still considered during computations. Auxin induces PINs and AUX/LAX protein expression (*Zwiewka et al., 2019*). PINs are subsequently delivered to the cell membranes according to mechanical constraints(AF), and one of the two polarization scenarios auxin-flux or regulator-polarizer, respectively.

All model components are presented in a comprehensive model diagram (*Figure 2—figure supplement 2* and *Figure 2—figure supplement 3*). Optimal parameters values were chosen by testing over a large plausible range of values for each parameter using high-throughput simulations on a computing cluster. Parameters description and values are listed in *Table 1*. Non-linearities of higher order used in some formulas simulate a threshold memory and signal amplification effects (increased sensitivity) that would result from multi-cascade signaling events: that is kinases and phosphatases such as MAPK (*O'Shaughnessy et al., 2011*).

## Anisotropy factor (AF) and cell polarity

The processes that define cell polarity in plants are not well understood (*Dettmer and Friml, 2011*), and are considered to be different from those in animals. Plant cells display clear polarity patterns when observed to grow anisotropically or by targeting proteins to specific regions in the cell membranes such as PIN proteins (*Yang, 2008*). Apart from PIN protein (*Wisniewska et al., 2006*), several other prominent cell polarity markers have been identified in plants, such as putative regulators of cell division orientation BASL (*Pillitteri et al., 2011*) and SOSEKI (*Yoshida et al., 2019*). So far, the only well-characterized regulators of PIN trafficking are the AGCVIII kinases and PP2A phosphatases (*Barbosa et al., 2018*), components of the phosphorylation on/off switch aimed at the central hydrophilic loop of PINs (*Michniewicz et al., 2007*). Cell polarity may be also influenced by a mechanical stimulus, nutrient availability, and pathogen responses (*Adamowski and Friml, 2015*).

Root cells present a clear apical-basal (shootward-rootward) polarity, which allows them to target hormones and other signaling molecules in specific directions (*Kleine-Vehn and Friml, 2008*). Almost all cell types in *A. thaliana* root display clear anisotropic geometries (*Baskin, 2005*) while microtubules orientation may correlate with PIN protein subcellular localizations (*Heisler et al., 2010*). It has been recently shown that external stress can affect internal microtubules organization and therefore guide the anisotropy and orientation of CMT arrays (*Hamant et al., 2019*). During root swelling the cells are growing isotropically but also undergo stretching in a direction determined by their geometry and their position inside the organ. In line with these observations, the AF aligns perpendicular to the cell wall deformation, hence enforcing anisotropic cell growth (*Hamant et al., 2008*).

Key model assumptions regarding mechanics of cell growth and polarity are listed below:

1. The anisotropy factor (AF) is internally represented as a vector perpendicular to the longest principal growth axis. The length of the AF vector can range from 0 to 1 indicating the degree of induced anisotropy.
2. AF reorientation is triggered only if a certain wall deformation threshold is reached.
3. After cell division daughter cells initially inherit the AF configuration from the mother cell.
4. AF reorientation is defined by the following formula:

$$\frac{d\overrightarrow{AF_{cell}}}{dt} = \overrightarrow{AF_{cell}} + R_{AF} \sum_{i}^{m} u\left(\overrightarrow{AF_{cell}}\right) \left| \left( \left( u\left(\overrightarrow{AF_{cell}}\right) \cdot u\left(\overrightarrow{mem_i}\right) \right) \epsilon_{mem_i} \right) \right| - d_{AF} \overrightarrow{AF_{cell}} \tag{1}$$

where, $\overrightarrow{AF_{cell}}$ is the AF vector inside the cell; $R_{AF}$ is AF reorientation rate; $m$ is the total number of membrane sections; $u(\overrightarrow{AF_{cell}})$ is the unit vector parallel to the AF vector; $u\left(mem_i\right)$ is the unit vector parallel to membrane section $mem_i$; $\epsilon_{mem_i}$ is the deformation rate (strain) of membrane section $mem_i$; $d_{AF}$ is AF decay rate. The dot "." symbol indicates the dot product between vectors.

## Auxin transport module description

Previously proposed models of auxin polar transport can be divided into two main classes: flux-based and concentration-based models (*van Berkel et al., 2013*; *Wabnik et al., 2011*). Briefly, flux-based canalization models assume that PIN proteins polarize according to the direction of auxin flux (*Alim and Frey, 2010*; *Feugier et al., 2005*; *Feugier and Iwasa, 2006*; *Fujita and Mochizuki, 2006*; *Mitchison, 1997*; *Stoma et al., 2008*). In concentration-based models, the cell can detect auxin concentrations

**Table 1.** Model parameters.

| Parameter | Description | Value | Unit |
|---|---|---|---|
| Mechanical model component | | | |
| $R_{AF}$ | AF reorientation rate | 0.02 | $h^{-1}$ |
| $D_{AF}$ | AF degradation rate | 0.01 | $h^{-1}$ |
| Auxin transport model component | | | |
| $b_{IAA}$ | basal auxin production rate | 0* | nM/h |
| $DI_{IAA}$ | auxin diffusion rate in the intercellular space | 1 | $\mu m^2/h$ |
| $d_{IAAb}$ | basal auxin degradation rate | 0.0125, *Perianez-Rodriguez et al., 2021* | nM/h |
| $d_{IAAMax}$ | maximum auxin degradation rate coefficient | 0.125 | $h^{-1}$ |
| $K_{IAAMax}$ | coefficient for half-max auxin degradation | 5 | nM |
| $K_{AUX1}$ | coefficient of auxin importing rate by AUX/LAX | 1 | $\mu m/h$ |
| $K_{PIN}$ | coefficient of auxin export rate by PIN | 1.4, *Mironova et al., 2010* | $\mu m/h$ |
| $b_{AUX1}$ | AUX/LAX basal expression | 1 | nM/h |
| $AUX1_{expr}$ | auxin-induced AUX/LAX maximal expression | 30 | nM/h |
| $AUX1_K$ | auxin-induced AUX/LAX half-max expression | 0.01 | nM |
| $AUX1_{tr}$ | AUX/LAX trafficking rate | 1 | $h^{-1}$ |
| $AUX1_{Max}$ | maximum concentration of AUX/LAX | 2 | nM |
| $AUX1_{MaxMem}$ | maximum concentration of AUX/LAX on membrane sections | 15 | nM |
| $d_{AUX1}$ | AUX/LAX degradation rate | 0.08 | $h^{-1}$ |
| $b_{PIN}$ | PIN basal expression | 0.2, *Mironova et al., 2010* | nM/h |
| $PIN_{expr}$ | auxin-induced PIN maximal expression | 50 | nM/h |
| $PIN_K$ | auxin-induced PIN half-max expression | 0.05 | nM |
| $PIN_{tr}$ | PIN trafficking rate | 1 | $h^{-1}$ |
| $PIN_{Max}$ | maximum PIN concentration inside the cell | 2 | nM |
| $PIN_{MaxMem}$ | the maximum concentration of PIN on membrane sections | 15 | nM |
| $d_{PIN}$ | PIN degradation rate | 0.08, *Mironova et al., 2010* | $h^{-1}$ |
| $d_{PINmax}$ | maximum PIN degradation rate on membranes | 0.8 | $h^{-1}$ |
| $kAF$ | coefficient for AF orientation contribution to PIN sensitivity | † | - |
| $kP$ | coefficient for auxin flow contribution to PIN sensitivity | 3 | - |

*Table 1 continued on next page*

*Table 1 continued*

| Parameter | Description | Value | Unit |
|---|---|---|---|
| kAFP | coefficient for interaction AF orientation+ auxin flow contribution to PIN sensitivity | 3 | - |
| kG | coefficient for cell geometry contribution to PIN sensitivity | 3 | - |
| Kaf | half-max AF orientation contribution to PIN sensitivity | 0.5 | - |
| Kgeom | half-max cell geometry contribution to PIN sensitivity | 0.5 | - |
| PIN polarization parameters | | | |
| $Kflux$ | auxin-flux half-max contribution on PIN sensitivity | 0.1 | nM μm |
| $b_{REG}$ | regulator basal expression | 10 | nM/h |
| $b_{POL}$ | polarizer basal expression | 10 | nM/h |
| $d_{POL}$ | regulator decay rate | 0.08 | h$^{-1}$ |
| $d_{REG}$ | polarizer decay rate | 0.08 | h$^{-1}$ |
| $Kreg_{tr}$ | regulator base trafficking rate | 1 | h$^{-1}$ |
| $Kpol_{tr}$ | polarizer base trafficking rate | 0.01 | h$^{-1}$ |
| $D_{reg}$ | regulator diffusion rate | 1 | μm$^2$/h |
| $D_{pol}$ | polarizer diffusion rate | 0.1 | μm$^2$/h |
| $Kdisp_{POL}$ | polarizer displacement rate | 10 | h$^{-1}$ |
| $Kreg_{IAA}$ | regulator auxin-induced half-max trafficking rate | 0.01 | nM |
| $Kpol_{IAA}$ | polarizer auxin-induced half-max trafficking rate | 0.01 | nM |
| $Kreg_{GradT}$ | regulator max trafficking rate activation by auxin gradient | 1 | nM/h |
| $Kreg_{GradK}$ | regulator auxin gradient-induced half-max trafficking rate | 1 | nM |
| $Kpol_{IP}$ | half-max value of polarizer contribution on PIN sensitivity | 0.1 | nM |
| Auxin-dependent root growth parameters | | | |
| $kE_{Max}$ | cell wall maximum stiffness | 1 | - |
| $K_{1auxin}$ | half-max cell wall relaxation coefficient by auxin | 0.05 | nM |
| $K_{2auxin}$ | half-max cell wall stiffening coefficient by auxin | 3 | nM |

*auxin basal expression is set to zero for the default wild type model. However, when local production in the QC is necessary, the value is set to 10.

†this parameter is set to 0 in the default model and included in the formulas only for completeness.

of a surrounding environment and increase PIN transport either against the gradient (*Jönsson et al., 2006*; *Merks et al., 2007*; *Newell et al., 2008*; *Smith et al., 2006*) or with the gradient (*Kramer, 2009*; *Wabnik et al., 2010*). Despite relying on different formulations, both types of models assume auxin feedback on PIN polarity which can recreate some aspects of auxin-related patterning observed in plant development. An alternative model was proposed by *Heisler et al., 2010*. The authors

suggested a correlation between PINs polarity and the alignment of cortical microtubules, indicative that the cell wall stress could be involved in determining PIN localizations. Interestingly, a more recent study (**Narasimhan et al., 2021**) showed that auxin exhibits a PIN2-specific positive effect on endocytosis, indicating a potential role for auxin in blocking PIN protein recruitment.

However, we primarily focused on the auxin-flux model and its molecular realization in this study. In our model, both PINs and AUX/LAX expressions are induced by the presence of auxin inside the cell (**Vieten et al., 2005**). Similarly, PIN trafficking is positively or negatively regulated by auxin depending on one of two scenarios (Auxin-flux module description and Regulator-Polarizer module description sections). Auxin is exported by PINs from the cell into an intercellular space where it can be imported by AUX1 that is uniformly distributed on the membranes. The set of model assumptions and components for auxin transport is listed below:

1. The cell membrane is represented by a two-dimensional polygon. Each edge of the polygon denotes a section of the cell wall/membrane (*mem*). Each membrane section stores mechanical and biochemical attributes, such as PIN and AUX/LAX levels, intercellular auxin concentration, and AF orientation. Note that amounts of auxin and transporters are given in concentrations; the number of molecules divided by the area of the cell or the intercellular space. For example, $IAA_{cell} = \frac{molecules\ of\ IAA}{area\ of\ the\ cell}$.

2. Auxin is imported by AUX/LAX from the intercellular space and exported in a polar manner from cells by PINs with the support of the PGP1/ABC transporter family (**Geisler and Murphy, 2006**). However, we do not include the PGP1/ABC transporters in the current model, therefore active auxin transport is expressed by the combined action of PIN and AUX/LAX carriers:

$$I_{mem} = K_{AUX1}\,AUX1_{mem}\,IAA_{mem}\,L_{mem} \tag{2}$$

$$E_{mem} = K_{PIN}\,PIN_{mem}\,IAA_{cell}\,L_{mem} \tag{3}$$

$$\frac{dIAA_{mem}}{dt} = \sum_i^n \left(E_{mem} - I_{mem}\right)_{cell_i} - \sum_i^m DI_{IAA}\frac{IAA_{mem} - IAA_{mem_i}}{L_{mem} + L_{mem_i}} - D_{IAA}\left(IAA_{cell}\right) - d_{IAA}\left(IAA_{mem}\right) \tag{4}$$

$$\frac{dIAA_{cell}}{dt} = b_{IAA} + \sum_i^m \left(I_{mem} - E_{mem}\right)_{cell_i} + D_{IAA}\left(IAA_{cell}\right) - d_{IAA}\left(IAA_{cell}\right) \tag{5}$$

where, *IAA*mem is the auxin imported into the cell through a specific membrane section *mem*; $K_{AUX1}$ is the auxin import rate of AUX/LAX; $AUX1_{mem}$ is the amount of AUX/LAX protein localized on membrane section *mem*, $IAA_{mem}$ is the intercellular auxin available in the membrane section *mem*. $E_{mem}$ is the auxin exported from the cell into the intercellular space through a specific membrane section *mem*; $K_{PIN}$ is the auxin export rate of PIN; $PIN_{mem}$ is the amount of PIN protein localized on membrane section *mem*, *IAA*cell is the concentration of auxin inside the cell; $L_{mem}$ is the length of membrane section *mem*. $IAA_{mem}$ is the intercellular auxin in the membrane section *mem*; $IAA_{mem_i}$ is the amount of intercellular auxin of a cell neighbor; $cell_i$ is a cell sharing the current membrane section *mem*; $mem_i$ is a membrane section neighboring the current membrane section *mem*; $DI_{IAA}$ is the auxin diffusion rate in the intercellular space; $L_{mem}$ is the length of membrane section *mem*; $L_{mem_i}$ is the length of the neighboring membrane section $mem_i$; $d_{IAA}$ is the auxin degradation in the current membrane section *mem*. $IAA_{cell}$ is the auxin concentration inside the *cell*; $b_{IAA}$ is the auxin basal production rate; $D_{IAA}(IAA_{cell})$ is the net auxin diffusive import into the *cell*; $d_{IAA}(IAA_{cell})$ is the auxin degradation for the current *cell*.

3. Auxin can diffuse passively into cells from the intercellular space (**Petrášek and Friml, 2009**) according to the formula:

$$D_{IAA}\left(IAA_{cell}\right) = \sum_i^m P_{IAA}\left(IAA_{mem_i} - IAA_{cell}\right)L_{mem_i} \tag{6}$$

where, $D_{IAA}(IAA_{cell})$ is the net auxin diffusive import between the cell and the surrounding intercellular space; *m* is the total number of membrane sections; $P_{IAA}$ is membrane permeability; $IAA_{mem_i}$ is the intercellular auxin in the membrane section $mem_i$; $IAA_{cell}$ is the auxin concentration inside the cell; $L_{mem_i}$ is the length of membrane section $mem_i$.

4. Auxin decay follows the combined effect of conjugation and oxidation (**Ljung, 2013**), at a constant rate in our model. If auxin inside the cell or a membrane section *mem* reaches a high auxin concentration threshold, auxin degradation is increased to balance the total auxin concentration. This is necessary to preclude excessive auxin levels that would retard root growth (**Fendrych et al., 2018**):

$$d_{IAA}\left(IAA_{cell/mem}\right) = d_{IAAb} + (d_{IAAMax} - d_{IAAb})\frac{IAA^4_{cell/mem}}{K^4_{IAAMax}+IAA^4_{cell/mem}} \tag{7}$$

where, $d_{IAA}\left(IAA_{cell/mem}\right)$ is the auxin degraded inside a cell or in the intercellular space; $d_{IAAb}$ is the basal auxin degradation rate; $d_{IAAMax}$ is the maximum auxin degradation rate; $IAA_{cell/mem}$ is the current auxin concentration inside the cell or in the membrane section *mem*, respectively; $K_{IAAMax}$ is the coefficient for half-max auxin degradation.

5.  Auxin regulates the amount of the auxin carriers (*Heisler et al., 2005*; *Vieten et al., 2005*), by increasing PINs and AUX/LAX expression:

$$\frac{dAUX1_{cell}}{dt} = b_{AUX1} + AUX1_{Expr}\frac{IAA^2_{cell}}{AUX1^2_K+IAA^2_{cell}} - AUX1_{cell}\,AUX1_{tr} - d_{AUX1}\,AUX1_{cell} \tag{8}$$

$$\frac{dPIN_{cell}}{dt} = b_{PIN} + PIN_{Expr}\frac{IAA^2_{cell}}{PIN^2_K+IAA^2_{cell}} - PIN_{cell}PIN_{tr} - d_{PIN}PIN_{cell} \tag{9}$$

where, $AUX1_{cell}$ is the cytoplasmic AUX/LAX pool; $b_{AUX1}$ is AUX/LAX basal expression; $AUX1_{Expr}$ is the auxin-induced AUX/LAX maximal expression; $AUX1_K$ is the auxin-induced AUX/LAX half-max expression; $IAA_{cell}$ is the auxin concentration inside the cell; $AUX1_{tr}$ is AUX/LAX trafficking rate; $d_{AUX1}$ is AUX/LAX degradation rate. AUX/LAX expression is limited by the maximum concentration $AUX1_{Max}$ (see *Table 1*). $PIN_{cell}$ is the cytoplasmic PIN pool; $b_{PIN}$ is the PIN basal expression; $PIN_{expr}$ is the auxin-induced maximal PIN expression; $PIN_K$ is the auxin-induced PIN half-max expression coefficient; $IAA_{cell}$ is the auxin concentration inside the cell; $PIN_{tr}$ is PIN trafficking rate; $d_{PIN}$ is PIN degradation rate. PIN expression is limited by the maximum concentration $PIN_{Max}$ (see *Table 1*).

6.  Auxin modulates the subcellular localization of PIN proteins in a feedback-dependent manner (*Sauer et al., 2006*). AUX/LAX is redistributed evenly among the membrane sections, while PINs are redistributed depending on the 'PIN sensitivity' of each membrane section:

$$\frac{dAUX1_{mem}}{dt} = AUX1_{cell}AUX1_{tr}\frac{L_{mem}}{\sum^m_i L_{mem_i}} - d_{AUX1}AUX1_{mem} \tag{10}$$

$$\frac{dPIN_{mem}}{dt} = PIN_{cell}PIN_{tr}PinS_{mem} - d_{PIN_{mem}}PIN_{mem} \tag{11}$$

$$d_{PIN_{mem}} = d_{PIN} + \left(d_{PINmax} - d_{PIN}\right)\frac{1}{1+IAA_{cell}} \tag{12}$$

where, $AUX1_{mem}$ are the AUX/LAX in the membrane section *mem*; $AUX1_{cell}$ is the concentration of AUX/LAX in the cytoplasm; $AUX1_{tr}$ is AUX/LAX trafficking rate; *m* is the total number of membrane sections; $L_{mem}$ is the length of membrane section *mem*; the element $\frac{L_{mem}}{\sum^m_i L_{mem_i}}$ therefore indicates the fraction of cytoplasmic AUX/LAX trafficked to the membrane section *mem*; $d_{AUX1}$ is AUX/LAX degradation rate. AUX/LAX trafficking is disabled when the membrane section is saturated $AUX1_{Maxmem}$ (see *Table 1*). $PIN_{mem}$ is the PIN proteins on membrane section *mem*; $PIN_{cell}$ is the concentration of PIN in the cytoplasm; $PIN_{tr}$ is PIN trafficking rate; $PinS_{mem}$ is the PINs sensitivity of membrane section *mem*, which determines the fraction of cytoplasmic PINs that are trafficked to membrane section *mem*; $d_{PIN_{mem}}$ is PIN degradation dynamic formula on the membranes, described in *Equation 12*. PIN trafficking is disabled when the membrane section is saturated $PIN_{Maxmem}$ (see *Table 1*). $d_{PINmem}$ is PIN degradation dynamic formula on the membranes; $d_{PIN}$ is PIN base degradation rate (same as cytoplasmic degradation); $d_{PINmax}$ is the maximum PIN degradation on the membranes; $IAA_{cell}$ is the current auxin concentration inside the cell; PIN degradation in the membrane is enhanced when auxin in the cell is low, to facilitate rapid PIN polarity reestablishment.

7.  PINs are trafficked to the membranes according to a specific criterion; namely, each membrane section *mem* possesses an instrumental 'PIN sensitivity' property, which regulates the propensity of that membrane section to incorporate additional PINs. This property is a phenomenological parameter for more low-level processes involved in PIN trafficking, such as PINs phosphorylation and endocytosis. PIN sensitivity of each membrane varies between 0 and 1 and the total PIN sensitivity of all membranes sections sum up to 1. PIN sensitivity is the linear combination of auxin flow (defined either by auxin flux or auxin concentrations), growth anisotropy (defined by AF), and lesser extent the cell geometry (*Elliott and Kirchhelle, 2020*). Columella cells are the only exception to this rule; in the columella, PINs are always trafficked uniformly among membrane sections to reflect the observed PIN3 distribution (*Friml et al., 2002*). PIN sensitivity of a given membrane section *mem* is defined as:

$$PinS_{mem} = \frac{exp(PinSR_{mem})}{\sum_i^m exp(PinSR_{mem_i})} \tag{13}$$

$$PinSR_{mem} = kAF\, IAF_{mem} + kP\, IP_{mem} + kAFP\, (IAF_{mem} + IP_{mem}) + kG\, IG_{mem} \tag{14}$$

$$IAF_{mem} = \left|\overrightarrow{AF_{cell}}\right| \frac{U\left(\overrightarrow{AF_{mem}}\right)^4}{U\left(\overrightarrow{AF_{mem}}\right)^4 + Kaf^4}; U\left(\overrightarrow{AF_{mem}}\right) = u\left(\overrightarrow{AF_{cell}}\right) \cdot n\left(\overrightarrow{mem}\right) \tag{15}$$

$$IG_{mem} = \left(u\left(\overrightarrow{axisMax}\right) \cdot n\left(\overrightarrow{mem}\right)\right) \frac{\left(1 - \frac{\left|\overrightarrow{axisMin}\right|}{\left|\overrightarrow{axisMax}\right|}\right)^4}{\left(1 - \frac{\left|\overrightarrow{axisMin}\right|}{\left|\overrightarrow{axisMax}\right|}\right)^4 + Kgeom^4}$$

where, PIN sensitivity of membrane section *mem* is obtained by applying the soft-max function over all the raw PIN sensitivities $PinSR_{mem_i}$ calculated for each membrane section of the cell. The soft-max function was used to normalize the total sum of PIN sensitivities to 1. $IAF_{mem}$ is the AF contribution to PIN sensitivity of membrane section *mem* (see the *Equation 15*); $kAF$ is the weight of AF contribution to PIN sensitivity; $IP_{mem}$ is the auxin flow contribution to PIN sensitivity of membrane section *mem* (this parameter depends on auxin-flux and regulator-polarizer models, see sections 1.7 and 1.8); $kP$ is the coefficient of auxin flow contribution to PIN sensitivity; $kAFP$ is the coefficient of combined action of the AF and auxin flow to PIN sensitivity; $IG_{mem}$ is the contribution of cell geometry to PIN sensitivity of membrane section *mem* (see the *Equation 16*); $kG$ is the coefficient of the contribution of cell geometry to PIN sensitivity. $IAF_{mem}$ describes AF impact on PIN sensitivity of membrane section *mem*. $\overrightarrow{AF_{cell}}$ is the AF vector for the cell (defined in *Equation. 1*); $U\left(\overrightarrow{AF_{mem}}\right)$ is the normalized effect of AF on membrane section *mem*; $u\left(\overrightarrow{AF_{cell}}\right)$ is the unit vector parallel to the AF vector; $n\left(\overrightarrow{mem}\right)$ is the unit vector orthogonal to membrane section *mem*. $Kaf$ half-max constant of AF effect on membrane PIN sensitivity. $IG_{mem}$ denotes the impact of cell geometry on PIN sensitivity of membrane section *mem*; $n\left(\overrightarrow{mem}\right)$ is the unit vector orthogonal to membrane section *mem*; $\overrightarrow{axisMax}$ and $\overrightarrow{axisMin}$ are the longest and the shortest cell principal axis, respectively; $u\left(\overrightarrow{axisMax}\right)$ is the unit vector parallel to the longest cell axis; $Kgeom$ is the half-max coefficient of cell geometry contribution to PIN sensitivity. The dot '.' symbol indicates the dot product between vectors.

As discussed before, the PIN sensitivity of a specific membrane section depends on whether the auxin flux or regulator-polarizer method is used (see sections 1.7 and 1.8). Either of these two scenarios determines the $IP_{mem}$ term in *Equation (14)*.

## Auxin-flux module description

Flux-based computer models of auxin transport were first introduced by *Mitchison, 1997*. A general mechanism is that cells sense the auxin flux and based on that information cells increase their auxin transport capacity in the flux direction. This mechanism reproduces canalized auxin transport patterns during leaf vein formation (*Rolland-Lagan and Prusinkiewicz, 2005*). Flux-based models assume the existence of cellular flux-sensing components that have not been yet experimentally identified. Using the auxin-flux model, cells recognize the net vector of auxin flow and redirect PINs accordingly. Specifically, the auxin-flux vector of a cell is defined as:

$$\overrightarrow{FLUX}_{cell} = \sum_i^m \overrightarrow{u\left(centroid_{cell}\ midpoint_{mem_i}\right)} \left(I_{mem_i} - E_{mem_i}\right) \tag{16}$$

where, *m* is the total number of membrane sections; $centroid_{cell}$ is the centroid of the cell, $midpoint_{mem_i}$ is the midpoint of membrane section *mem*; $u\left(centroid_{cell}, midpoint_{mem_i}\right)$ is the unit vector parallel to the vector connecting the two previous elements; $\left(I_{mem_i} - E_{mem_i}\right)$ indicates the net amount of auxin crossing the membrane section *mem*.

Given the auxin-flux vector of a cell, the contribution of auxin flow to PIN sensitivity of a membrane section *mem* is obtained by projecting the auxin flux over the membrane section *mem*:

$$IP_{mem} = \frac{F_{mem}^4}{F_{mem}^4 + Kflux^4}; F_{mem} = \left(\overrightarrow{FLUX_{cell}} \cdot \overrightarrow{n\,(mem)}\right) L_{mem} \tag{17}$$

here, $IP_{mem}$ is the (unitless) auxin flow contribution to PIN sensitivity of a membrane section $mem$; $F_{mem}$ is the effect of the auxin flux on membrane section $mem$; $Kflux$ is the flux sensing constant; $\overrightarrow{FLUX_{cell}}$ is the auxin flux vector; $\overrightarrow{n\,(mem)}$ is the unit vector orthogonal to the membranes section $mem$ and $L_{mem}$ is the length of the membrane section $mem$. The dot "." symbol indicates the dot product between vectors.

## Regulator-polarizer module description

Cell polarization has been investigated both on the theoretical ground (*Gierer and Meinhardt, 1972*; *Jilkine and Edelstein-Keshet, 2011*; *Meinhardt and Gierer, 2000*) and designed synthetic circuits (*Chau et al., 2012*; *Rappel and Edelstein-Keshet, 2017*). We propose a mechanistic realization of auxin flux sensing by combining the interaction between four molecules: PIN, auxin, a polarizer, and a regulator. The polarizer is a molecule that promotes the sorting of PINs to the membrane section where it is most abundant (i.e. a specific kinase that phosphorylates PIN). The regulator is a molecule that is activated by auxin and inhibits polarizer abundance on the membranes (i.e. antagonizing phosphatase). Auxin presence in a membranes section promotes regulator trafficking, which in turn reduces the presence of the polarizer in that region. Free diffusion of the regulator over the surface of the cell results in the clustering of the polarizer on the opposite side of the cell where it promotes PIN trafficking by tuning the auxin contribution parameter $IP_{mem}$ (see *Equation. 14*).

- Auxin import across the plasma membrane is detected by the cell to promote regulator binding to the membrane. Specifically, the auxin influx-efflux ratio for each specific membrane section is used to determine the trafficking of the regulator:

$$Grad_{mem} = \left(\left(I_{mem} - E_{mem}\right) + \sum_i^m \frac{\left(I_{mem_i} - E_{mem_i}\right)}{distance\left(mem, mem_i\right)}\right) \frac{1000}{A_{cell}} \tag{18}$$

where $Grad_{mem}$ is the auxin influx-efflux ratio specific to membrane section $mem$; $\left(I_{mem} - E_{mem}\right)$ indicates the net amount of auxin crossing the membrane section $mem$; $distance\left(mem, mem_i\right)$ is the distance of membrane section $mem_i$ from our reference membrane section $mem$, calculated as the Euclidean distance between the two sections midpoints; $A_{cell}$ is the area of the cell. The metric is further normalized dividing by the cell area and amplified by an amplification factor of 1000.

- Regulator and polarizer are expressed and degraded at a constant rate in the cytoplasm. The trafficking of regulator and polarizer to the membranes is promoted by intracellular auxin:

$$\frac{dREG_{cell}}{dt} = b_{REG} - REG_{cell}\left(Kreg_{tr} + Kreg_{GradT}\frac{Grad_{mem}^4}{Grad_{mem}^4 + Kreg_{GradK}^4}\right)\frac{IAA_{cell}^2}{IAA_{cell}^2 + Kreg_{IAA}^2}$$
$$-d_{REG}REG_{cell} \tag{19}$$

$$\frac{dPOL_{cell}}{dt} = b_{POL} - POL_{cell}\,Kpol_{tr}\frac{IAA_{cell}^2}{IAA_{cell}^2 + Kpol_{IAA}^2} - d_{POL}\,POL_{cell} \tag{20}$$

where, $REG_{cell}$ and $POL_{cell}$ are the regulator and polarizer concentrations inside the cell, respectively; $b_{REG}$ and $b_{POL}$ are the regulator and polarizer basal production rate, respectively; $Kreg_{tr}$ and $Kpol_{tr}$ are the regulator and polarizer trafficking rates; $Kreg_{GradT}$ and $Kreg_{GradK}$ are regulator maximum trafficking rate and trafficking constant, respectively; $Grad_{mem}$ is auxin influx-efflux ratio for each specific membrane section $mem$; $IAA_{cell}$ is the amount of auxin inside the cell; $Kreg_{IAA}$ and $Kpol_{IAA}$ are regulator and polarizer trafficking constants, respectively; $d_{REG}$ and $d_{POL}$ are the regulator and polarizer degradation rates.

- The changes of regulator and polarizer species present on a specific membrane section $mem$ are defined by the following equations:

$$\frac{dREG_{mem}}{dt} = REG_{cell}\left(Kreg_{tr} + Kreg_{GradT}\frac{Grad_{mem}^4}{Grad_{mem}^4 + Kreg_{GradK}^4}\right)\frac{L_{mem}}{\sum_i^m L_{mem_i}}$$
$$\frac{IAA_{cell}^2}{IAA_{cell}^2 + Kreg_{IAA}^2} + D(REG_{mem} - d_{REG}REG_{mem}) \tag{21}$$

$$\frac{dPOL_{mem}}{dt} = POL_{cell}\, Kpol_{tr}\, \frac{L_{mem}}{\sum_i^m L_{mem}}\, \frac{IAA_{cell}^2}{IAA_{cell}^2 + Kpol_{IAA}^2}$$
$$+ D(POL_{mem}) + G(POL_{mem}) - d_{POL} POL_{mem} \tag{22}$$

where, $REG_{mem}$ and $POL_{mem}$ the regulator and polarizer in the membrane section *mem*. $Kreg_{tr}$ and $Kpol_{tr}$ are the regulator and polarizer trafficking rates, respectively; $REG_{cell}$ and $POL_{cell}$ are cytoplasmic pools; $Kreg_{IAA}$ and $Kpol_{IAA}$ are the regulator and polarizer trafficking constants; $L_{mem}$ is the length of the membrane section *mem*; $IAA_{cell}$ is the concentration of auxin inside the cell; $D(REG_{mem})$ and $D(POL_{mem})$ diffusion terms for regulator and polarizer, respectively; $G(POL_{mem})$ is the net fraction of polarizer displaced by the regulator in the membrane section *mem*.

- Regulator and polarizer diffuse along the cell membrane according to the following equations:

$$D(REG_{mem}) = D_{REG} \sum_i^{mem \pm 1} (REG_{mem_i} - REG_{mem}) \tag{23}$$

$$D(POL_{mem}) = D_{POL} \sum_i^{mem \pm 1} (POL_{mem_i} - POL_{mem}) \tag{24}$$

where, $D_{REG}$ and $D_{POL}$ are the regulator and polarizer diffusion rates, respectively; $REG_{mem}$ and $POL_{mem}$ are the regulator and polarizer in the membrane section *mem*, respectively; The polarizer is displaced by the presence of regulator molecules toward the zone where the concentration of the regulator is the lowest. To simulate this process, we apply the model of stochastic recruitment of molecules to the membrane sections (*Chau et al., 2012*): in a membrane section *mem*, a fixed batch of a polarizer is reserved for the displacement; then one of the adjacent membrane segments is selected randomly; if the adjacent segment contains less regulator than the current segment, the batch of a polarizer is moved to that neighboring segment. Polarizer displacement can be written as:

$$G(POL_{mem}) = Kdisp_{POL}\ if\ (REG_{mem+i} > REG_{mem}\ then\ POL_{mem+i}\ else\ 0 \tag{25}$$

$$i = random\ (-1, +1)$$

where $G(POL_{mem})$, is the amount of polarizer displaced by the regulator and $Kdisp_{POL}$ is the polarizer displacement rate.

- Given the calculated amount of polarizer in a given membrane segment, the auxin flow contribution term $IP_{mem}$ to PIN sensitivity (see *Equation. 14*) becomes:

$$IP_{mem} = \frac{POL_{mem}^4}{POL_{mem}^4 + Kpol_{IP}^4} \tag{26}$$

where, $Kpol_{IP}$ is the half-max constant.

## Cell growth description

The classical morphogen gradient model dictates that the cell fate is regulated by the positional information encoded in different morphogen levels at different positions across tissue (*Wolpert, 1969*). However, in an expanding system, cells are displaced quickly enough along the tissue experiencing different effective morphogens concentrations that depend on their current distance for the morphogens source(s). Moreover, cell growth dilutes morphogens concentration and thus modulates the morphogen effect on cell signaling. Our model addresses these issues by monitoring the combined effect of cell growth and auxin concentration on root development. The root growth component includes the following assumptions:

1. Cells expand according to the chemiosmotic theory of auxin transport (*Rayle and Cleland, 1992*). Cells are under constant turgor pressure which is resisted by the elastic effect of cell walls. In the model, the osmotic pressure is simulated by a PBD constraint described in Position-based dynamics implementation.
2. Increasing auxin concentration induces the relaxation of the cell walls allowing cell expansion under turgor pressure. On the contrary, high auxin levels disable cell walls relaxation (*Fendrych et al., 2018*). The relationship connecting cell walls stiffness and auxin is expressed by the following formula:

$$kE_{wall} = kE_{Max}\left(\frac{K_{1IAA}^4}{IAA_{cell}^4 + K_{1IAA}^4} + \frac{IAA_{cell}^4}{IAA_{cell}^4 + K_{2IAA}^4}\right) \tag{27}$$

$kE_{wall}$ is the extensional stiffness of the cell wall; $kE_{Max}$ is the maximum stiffness a cell wall can achieve; $K_{1IAA}$ is the auxin-induced cell wall relaxation coefficient; $K_{2IAA}$ is the auxin-induced cell wall stiffening coefficient; $IAA_{cell}$ is the auxin concentration inside the cell.

3. Cell growth is directionally constrained by the action of cellulose microfibrils that control anisotropic growth (*Baskin, 2005*). In the model, the action of cellulose microfibrils is simulated by the AF, which results from the action of a specific strain-based constraint (see Position-based dynamics implementation). For a given cell, the PBD strain constrains the stiffness of the membrane/wall segment depending on the alignment between the AF vector and this membrane/cell wall segment.

4. Cell division occurs when a certain area threshold is reached. The cell division plane passes through the centroid of the cell polygon and is parallel to the AF vector. There are however exceptions to this general rule, which are justified by experimental observation:

   a. Cortical/Endodermis Initials Daughters (CEID) cells always divide along the AF vector, to generate one cell of the same type and one CEID cell (*Miyashima and Nakajima, 2011*; *Mylona et al., 2002*).

   b. Cortical/Endodermis Initials Daughters (CEID) divide asymmetrically to generate one endodermal cell and one cortical cell (*Miyashima and Nakajima, 2011*; *Mylona et al., 2002*).

   c. Epidermis/Lateral Root Cap initials alternatively divide either orthogonal and parallel to AF vector to produce lateral root cap cells or epidermal cells, respectively (*Kumpf and Nowack, 2015*).

   d. Quiescent cells remain fixed (*Rovere et al., 2016*), and therefore we assume that these cells never grow or divide.

   e. Columella initials divide asymmetrically to generate one cell of the same type and one columella cell (*Scheres et al., 2002*). Columella cells are considered to be differentiated (*Kumpf and Nowack, 2015*).

   f. Vascular initials divide asymmetrically to generate one cell of the same type and one vascular cell (*Baum et al., 2002*).

## Position-based dynamics implementation

The typical approach to simulate dynamic growing systems in biology is based on force-based calculations (*Nealen et al., 2006*). Tissues are usually represented as triangulated meshes made of connected vertices and forces are accumulated on these vertices following specific biological criteria such as internal turgor pressure, anisotropic expansion, or gravity. Vertex acceleration is later derived from these forces and vertex masses according to Newton's second law. A time integration scheme is then used to first compute the velocities from the accelerations and then the final positions from the velocities. Classical integration methods are usually unstable or very computationally expensive, resulting in either unmanageable or extremely inefficient simulations. Therefore, instead of a forced-based system, we decided to implement the mechanical growth of *Arabidopsis thaliana* root using Position-Based Dynamics (PBD) (*Müller et al., 2007*). PBD is a recent method used to simulate physical phenomena such as cloth, deformation, fluids, fractures, material rigidity (*Müller et al., 2007*). PBD omits the velocity layer and instead computes the future positions of vertices based on mechanical constraints that restrict the system dynamics. The main PBD loop is summarized in the following diagram:

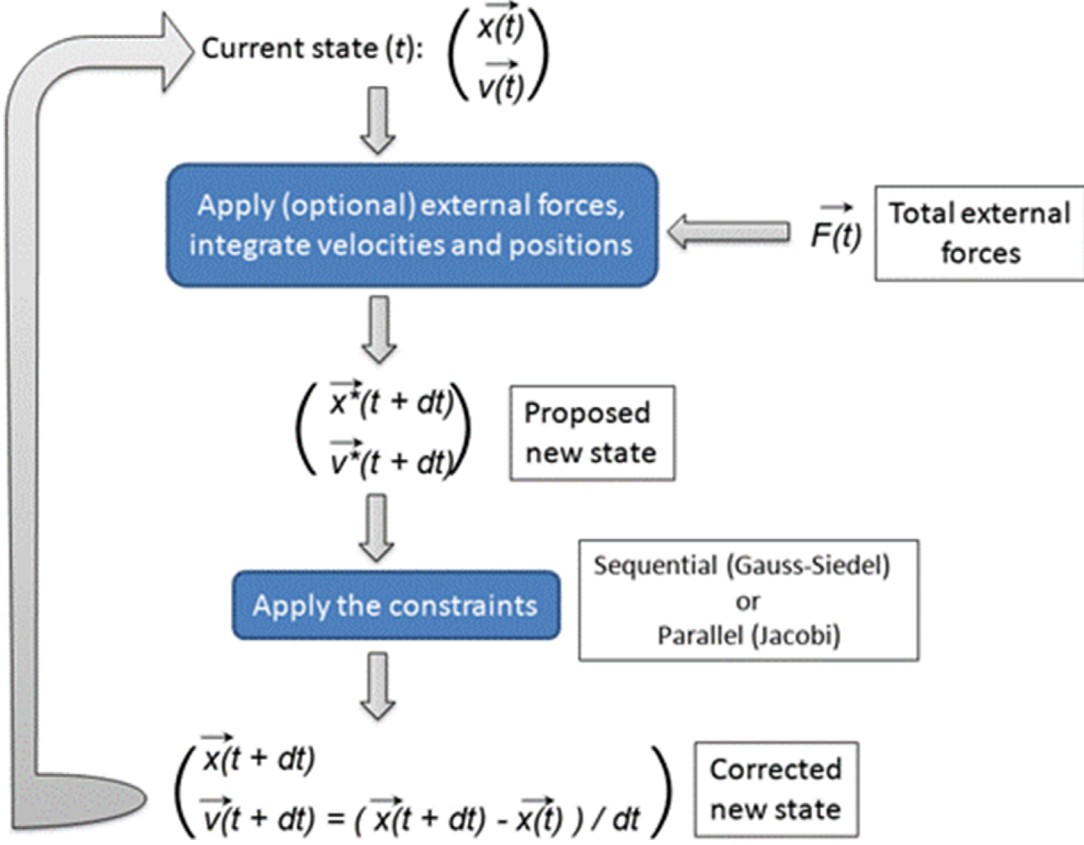

**Scheme 2.** Position-based dynamics algorithm.

The generic algorithm for PBD is described as following is pseudo-code:

```
// The algorithm assumes that each vertex in the model possesses the
following
// attributes: velocity, position and mass.
// Vertices velocities are updated by applying the external forces
(according to
// Newton's law) and it proposes new positions for each vertex.
for Vertex v in Vertices {
    v.previous_position = v.position;
    v.velocity += dt * (v.forces / v.mass);
    v.position += dt * v.velocity;
}

// The iterative solver applies the constraints to the proposed vertices
// positions, adjusting them such that they satisfy the constraints.
for i in iterations
    project_constraints(contraints, vertices);

// Finally, adjust vertices velocity based on the new adjusted positions
for Vertex v in Vertices
    v.velocity = (v.position – v.previous_position) / dt;
```

To provide a simple example of PBD constraint, consider the typical mass-spring system where two masses (usually represented by mesh vertices) are connected by an elastic spring. The elasticity of the spring applies a force on the masses which induces acceleration and velocity. In the PBD formulation, the same is achieved by projecting the distance constraint: $C(\mathbf{p_1}, \mathbf{p_2}) = |\mathbf{p_1} - \mathbf{p_2}| - d$; $\mathbf{p_1}$ and $\mathbf{p_2}$ are the vertices positions and $d$ is the spring resting length. The resulting corrections $\Delta\mathbf{p_i}$ are subsequently weighted according to the inverse masses $\mathbf{w_i} = 1/\mathbf{m_i}$. Finally, to account for the non-linear effect of the stiffness correction and to make it independent from the number of algorithm iterations, the stiffness correction is multiplied by $k' = 1 - (1 - k)^{1/n}$, as described in section 3.3 of the original paper (*Müller et al., 2007*). PBD has been implemented in our model framework based on the original method described in *Müller et al., 2007*.

The current model integrates distance, shape, strain, bending as well as pressure constraints (*Figure 1—figure supplement 2A,B*):

1. The distance constraint controls the distance between connected vertices (*Müller et al., 2007*). This constraint simulates the elastic and plastic behavior of cell walls. Cell edges can be either internal to the cell (representing the internal cytoskeleton and pectin matrix) or on the border (representing the cell walls). Edges have different stiffness for extension and compression. Cells are generally regarded as incompressible objects and therefore maximum compression stiffness is considered. Internal edges are meant to represent the cell pectin matrix and implemented as a viscoelastic material. Border edges represent the cell walls and are very stiff to prevent cell swelling from turgor pressure but can be relaxed in an auxin-dependent manner.
2. The shape constraint (*Müller et al., 2007*) simulates the mechanical forces involved in the preservation of the cytoskeleton. The shape constraint prevents cell deformation and collapses under external forces while allowing cell growth under internal pressure.
3. The strain constraint reproduces anisotropic growth (*Müller et al., 2014*). This constraint is applied on mesh triangles restricting wall deformation along the AF vector.
4. The pressure constraint (*Müller et al., 2007*) mimics the osmotic pressure inside the cell. This constraint allows isotropic cell expansion, which can be opposed by cell wall stiffness and restricted by the strain constraint (depending on the AF vector). This constraint also implements a time-dependent version of Position-Based Dynamics, called XPBD (*Macklin et al., 2016*).
5. The bending constraint prevents cell walls angle to drift too far away from the resting condition (*Müller et al., 2007*), hence avoiding cell collapse at the cytoskeleton resting state.

## Additional parameters sensitivity analysis

Our model was put to the test by varying two important parameters using high through model simulations on a computing cluster.

PINs trafficking to a specific membrane section of the cell is determined by the joint interaction between auxin flow and the AF. Briefly, the AF restricts PINs poles, whereas auxin flow discriminates between the cell poles. Parameter $kP$ regulates the strength of auxin flow contribution to PIN trafficking (*Equation. 14*). We varied this parameter (the default value in the wild-type simulation was set to 3) over a range of values (*Figure 5—figure supplement 3A*). In most cases, models were robust, unless this parameter was set to zero ($kP = 0$). In this scenario, auxin distribution is notably reduced compared to the wild-type situation ($kP = 3$), and auxin barely reaches tissues far from the QC (after refluxing back from the tip). Also, internal tissues that are usually replenished through lateralization are almost deprived of auxin (cortex and endodermis) (*Figure 5—figure supplement 3A*). On the contrary, by setting higher values ($kP \geq 5$) the predicted auxin flow was much stronger (*Figure 5—figure supplement 3A*) and the auxin-reflux loop induced by auxin lateralization from the epidermis into the cortex was increased, creating zones of auxin accumulation in the reflux region, as well as higher auxin accumulation in the pericycle (*Figure 5—figure supplement 3A*). These findings indicate the important role of auxin flow in PIN polarity determination, local auxin distribution, and therefore root growth.

Auxin induces cell wall relaxation according to *Equation (27)*. The relationship between auxin and cell stiffness is regulated by $K_{1auxin}$, - the auxin-induced cell wall relaxation coefficient. This parameter regulates wall stiffness response to changes in local auxin concentrations. Therefore, we simulated root growth by setting the coefficient $K_{1auxin}$ (the default value in the wild-type simulation is 0.05) over a range of parameter values (*Figure 5—figure supplement 3B*). The model was able to reproduce

correct root growth patterns, demonstrating the general robustness of the model against the selection of this parameter (*Figure 5—figure supplement 3B*). Low values of $K_{1auxin}$ do not seem to produce any visible alternations of the default root growth configuration, while much higher values reduce root growth (*Figure 5—figure supplement 3B*). Future modifications of our model could account for mechanistic components regulating this term, such as auxin-regulated enzymatic processes involved in the cell wall relaxation.

## Acknowledgements

We are grateful Richard Smith, Anne-Lise Routier, Crisanto Gutierrez and Juergen Kleine-Vehn for providing critical comments on the manuscript. Funding: This work was supported by the Programa de Atraccion de Talento 2017 (Comunidad de Madrid, 2017-T1/BIO-5654 to KW), Severo Ochoa (SO) Programme for Centres of Excellence in R&D from the Agencia Estatal de Investigacion of Spain (grant SEV-2016–0672 (2017–2021) to KW via the CBGP). In the frame of SEV-2016–0672 funding MM is supported with a postdoctoral contract. KW was supported by Programa Estatal de Generacion del Conocimiento y Fortalecimiento Científico y Tecnologico del Sistema de I + D + I 2019 (PGC2018-093387-A-I00) from MICIU (to KW). MG is recipient of an IST Interdisciplinary Project (IC1022IPC03).

## Additional information

### Funding

| Funder | Grant reference number | Author |
|---|---|---|
| Comunidad de Madrid | 2017-T1/BIO-5654 | Krzysztof Wabnik |
| Ministerio de Ciencia, Innovación y Universidades | PGC2018-093387-A-I00 | Krzysztof Wabnik |
| Ministerio de Ciencia, Innovación y Universidades | SEV-2016-0672 (2017-2021) | Marco Marconi Krzysztof Wabnik |
| Oregon State University | IC1022IPC03 | Marcal Gallemi |

The funders had no role in study design, data collection and interpretation, or the decision to submit the work for publication.

### Author contributions

Marco Marconi, Conceptualization, Data curation, Formal analysis, Investigation, MM designed the computer model and analysed data and wrote the manuscript, Methodology, Resources, Software, Validation, Visualization, Writing - original draft, Writing - review and editing; Marcal Gallemi, Formal analysis, Investigation, Methodology; Eva Benkova, Funding acquisition, Resources, Supervision; Krzysztof Wabnik, Conceptualization, Formal analysis, Funding acquisition, Investigation, Project administration, Resources, Supervision, Writing - original draft, Writing - review and editing

### Author ORCIDs
Marco Marconi ![ORCID] http://orcid.org/0000-0002-3457-1384
Krzysztof Wabnik ![ORCID] http://orcid.org/0000-0001-7263-0560

### Decision letter and Author response
Decision letter https://doi.org/10.7554/eLife.72132.sa1
Author response https://doi.org/10.7554/eLife.72132.sa2

## Additional files

### Supplementary files
• Transparent reporting form

## Data availability

All data generated or analysed during this study are included in the manuscript and supporting files. The source data for Figure 1D, Figure 2H-K, Figure 2-figure supplement 4, Figure 2-figure supplement 5H-K, Figure 3E,F, Figure 4C, Figure 4-figure supplement 1, Figure 5A-E and Figure 5-figure supplement 1A-C are provided in corresponding source data files. The computer model code and PBD implementation can be found here: https://github.com/PDLABCBGP/ROOTMODEL-PBD (copy archived at swh:1:rev:3251ec9fb61c1d726b2960195e15f74fe2dd9249). We received a copy of MorphoDynamX from Dr. Richard S. Smith, JIC, UK. To request MorphoDynamX source code please contact Dr. Smith directly via email Richard.Smith@jic.ac.uk.

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
