## [Editor Report]

The authors have created the first detailed model combining the mechanics of root growth with the dynamic regulation of auxin transport and patterning. Their novel model is capable of explaining the anisotropic longitudinal growth of plant roots and the complicated patterns of polarized auxin transport underlying auxin patterning.

---

## [Decision Letter]

**Decision letter after peer review:**

Thank you for submitting your article "A coupled mechano-biochemical framework for root meristem morphogenesis" for consideration by *eLife*. Your article has been reviewed by 3 peer reviewers, one of whom is a member of our Board of Reviewing Editors, and the evaluation has been overseen by Detlef Weigel as the Senior Editor. The following individual involved in review of your submission has agreed to reveal their identity: Victoria Mironova (Reviewer #3).

While all 3 reviewers highly value the novel biochemical-mechanical modeling framework developed and its use for studying plant root development and beyond, they all 3 raised major concerns regarding the claims that are made, the biological correctness of some of the model assumptions, and the clarity and transparancy with which modeling assumptions and limitations are being discussed.

Essential revisions:

1) The authors are required to more extensively discuss modeling details, assumptions and limitations. PDB methodology, how cytoskeleton and how PIN polarisation are modeled should get more attention in the main text. Also, the PIN patterning that is claimed to arise in a self-organized manner is clearly driven by the imposed auxin sources and sinks. Similarly, growth anisotropy derives from the growth asymmetry imposed, so it is not self-organized but has a clear directional source of information. This matters should be discussed explicitly.

2) In line with the above, the authors should tone down their conclusions regarding the self-organized nature of the patterning they observe, and make explicit what inputs are needed to get the patterning they observe. This requires adjustment of the title.

3) The authors should carefully check the biological correctness of the assumptions. As one reviewer pointed out the assumed relationships between microtubules, trafficking and PIN polarity are incorrect.

Reviewer #1 (Recommendations for the authors):

1. line 47 on first mention of growth anisotropy it should be explained what exactly is meant for readers not familiar with the subject.

2. Lines 67-70: The authors couple symmetry breaking and anisotropic growth with polar auxin transport. It should be clarified to what extent the experimental evidence supports this coupling and to what extent it is an assumption.

3. Line 97 a bit more explanation of what PBD does would be nice, remains rather vague/abstract here. i.e. what do the constraints do, is there some Hamiltonian incorporating these constraints that should be minimized? Something else?

4. It should be clarified whether simulations shown in Figure 1A and B contain auxin dynamics (I suppose not?) or only include tissue mechanics. Also, the authors should adjust the depiction of CMT orientation, even at 400% zoom the arrows drawn now are unfortunately hardly visible. Finally, here and for all later simulations it should be clarified in the legend for how long simulations were run.

5. Figure 1D: on the y-axis it states "growth increase per hour", what exactly is meant here

a growth rate increase, or a size increase? Please clarify.

6. CTM should read CMT at line 138/146/150?

7. Presumably at some point also the hypocotyl growth speed will increase. Is the model capable of sustaining root growth anisotropy if after an initial period of growth rate differences between radicle and adjoining tissue the adjoining tissue starts to grow at a similar speed? How does this depend on auxin dynamics?

8. Around line 145 a short mention of the relevance of intramembrane PIN diffusion and PIN internalization would be in place; anisotropic delivery of proteins only results in anisotropic patterns if diffusion is limited and/or proteins are continuously internalized and redeposited, a matter on which the last author has previously done excellent work.

9. Starting at lines 154 the authors compare two methods for auxin-driven PIN polarization. However, the second method, the so-called regulator polarizer mechanism, functions essentially the same as the first, the with-the-flux method, in that both promote PIN deposition at the location of highest auxin flux. The authors should explain why they compare these two PIN polarization mechanisms, what are their essential differences (i.e. why do the mechanisms give different results predominantly for lateral PIN orientations) , and why they did not investigate another frequently investigated PIN polarization mechanism generally referred to as up-the-gradient polarization. The way it is presented now is confusing, as less technical readers may get the wrong impression that it is with the flux and up the gradient type mechanisms that are compared.

Secondly, as the authors state, the regulator-polarizer mechanism is essentially a Turing-type reaction diffusion patterning mechanism within the cell, which typically have a characteristic parameter dependent wavelength. How does such a characteristic wavelength impact the potential of this mechanism to correctly polarize cells of different sizes as occur in the modeled root tip?

10. The authors state they implement an apolar AUX/LAX pattern for cells, however it is unclear whether this pattern is applied for all cells or only for particular cell types, as has often been the case in other studies. Please clarify.

11. Also, in light of different modeling approaches in the field, the authors should clarify in their main text if their model allows for a single intracellular and intrawall auxin concentration or rather allows for intracellular and intrawall concentration gradients. Additionally they should clarify in the main text whether it is intra or extracellular auxin concentrations governing cell growth. Finally, if it is a single intracellular auxin concentration that governs growth, does this not imply that auxin and PIN patterning merely impact growth rate of cells and not growth anisotropy? Please clarify.

12. The article claims to explain the self-organized growth and patterning of the plant root tip. However in the model the authors impose an auxin source in the middle vascular files and an auxin sink in the topmost outer cell files, thus quite a bit of auxin-related prepatterning is superimposed. Indeed, in a sense the polarity of auxin transport outside the modeled root tip domain was superimposed and the modeled domain should merely align accordingly, rather then truly self-organize its PIN patterning. The authors should adress this issue more explicitly.

The authors could test whether imposing either only vascular influx or top epidermal sinks or only a transient signal would suffice to induce correct PIN patterning (I presume that in Figure 3 the auxin source is only removed after the PIN pattern was established, not from the beginning onwards) ? Even more important, the authors should test to what extent mechanical feedback is necessary for the observed PIN patterning given the imposed auxin source and sink locations and auxin-PIN feedback mechanisms (of course in these simulations PIN membrane delivery should be made independent from CMT orientation). Finally, in cells with lateral inwards PIN, is CMT organization also less longitudinally and more diagonally or even transversally oriented, and if so how does this arise from tissue growth mechanics? Please clarify.

13. line 206, please specify which type/subset of parameters could be fitted based on these data.

14. line 210, please specify the nature of the peak (i.e. a peak of what).

15. For Figure 2H-K please clarify whether the shown 1D profiles are for a specific cell file, or rather an average across all cell files for that particular position along the longitudinal axis.

16. Line 214 please consider reformulating "eventually reproducing the non-trivial shape of the root". The model reproduces growth direction anisotropy and PIN and auxin patterning, yet as far as I can judge does not provide an explanation for the wedge shape of the root tip, or the precise organization of the SCN and the differentially oriented divisions occurring there or the precise number of cell files present in the root tip.

17. Line 238 Please clarify the "plausible range of parameter values" statement. Over what range were parameters varied, which parameters?

18. Line 239 please consider reformulating "prediction" into "emergent property", as the phenomenon has long been known it can hardly be seen as a model prediction, at the same time that it automatically arises from the model as an emergent property does deserve more attention.

Also, the authors could possibly test the explanation they offer for the bidirectional auxin flow in the cortex by artificially manipulating epidermis/lateral root cap auxin flow and seeing how this affects cortical PIN patterning.

19. In Figure 3C it appears as if -compared to Figure 3D- PIN proteins are not or hardly present on the membrane. Please clarify and explicitly discuss this matter in the text.

20. Auxin levels in Figure 2C/E-G and Figure 5A in particularly the vasculature seem much lower than in Figure 3C/D. Please clarify?

21. In Figure 3E and F, where in the root are growth rate or auxin concentration measured. Please clarify in figure legend.

22. Figure 5D, the cartoon to the left suggests that in the root tip cut experiment only downward PIN mediated auxin flow occurs, please clarify if this is an emergent result from the root tip cut or rather that PIN2 mediated flow has been abolished.

23. Figure 5E, please clarify whether the narrowness of the root at the top is a result of imposed mechanical boundary conditions.

24. Line 354 Please replace predict with reproduce.

25. In the discussion the authors should be a bit more modest in their statements on the models ability to reassemble key root properties given the importance of superimposed mechanical asymmetries, auxin sources and sinks as discussed earlier.

26. The authors should describe in the main text in a bit more detail how exactly does auxin translate into cellular growth rate and how does this ensure stable, coordinated growth across cell files. In a previous study it was shown that since auxin levels differ significantly across cell files (e.g. much higher in vasculature than in neighboring cell files), problems in coordinated cell growth may occur (https://pubmed.ncbi.nlm.nih.gov/25358093/). Why is that not happening here. Please explain.

27. In the discussion authors mention possible uses of the model such as studying tropisms. However the latter requires incorporating an elongation zone in which cells undergo rapid and extreme cell elongation. It seems that the current model only incorporates slow cytoplasmic cell growth and division occurring in the meristem, would the used model formalism be capable of describing rapid cell elongation? Would the model formalism be capable of simulating the growth asymmetries occurring in root tropisms?

28. The simulation code is made available to reviewers, Will the code be made publicly available upon publication?

29. Equation 4 and descriptions thereof: would it not make more sense to denote this as apoplastic rather than membrane auxin levels (although I realize apoplast and membrane are a single entity in the model formalism). Also the subscript in the first term should not read cell i but membrane/apoplast i I believe.

30. Equation 7, the authors state that auxin decay rate increases beyond a certain threshold auxin level, they should add on what experimental data this is based.

31. Equations 8 and 9: what do the terms AUX1_cell*AUX1_tr and PIN_cell*PIN_tr mean? Only later clear it is about trafficking, explain at first occurrence please.

32. The model contains quite a lot of non-linear interactions, with particularly for the flux and regulator-polarizer model powers of 4, the authors should clarify if model outcomes critically depend on these strong non-linearities.

General comment: English grammar is quite poor, requires correction.

*Reviewer #2 (Recommendations for the authors):*

Some points would require attention:

1) Misconception on the role of cytoplasmic microtubules: "CTMs restrict the deposition of various protein cargoes on the plasma membrane, typically along the maximal growth direction (maximal strain) (Adamowski et al., 2019; Nieuwland et al., 2016; Siegrist and Doe, 2007; Yang, 2008). It is plausible that PIN protein allocation (and/or other cargoes) at the plasma membrane might be restricted by CTMs." I have a problem with this claim. In Heisler 2010, it was shown that PIN1 remains polarly localized when microtubules are depolymerized. How does that fit with this model? Wouldn't it make more sense to involve the link between CESA and PIN1 (Feraru), or membrane tension and vesicle trafficking (Nakayama 2012, Heisler 2010)? I think the authors rather want to say that mechanical stress is vectorial in essence, and thus could guide both growth anisotropy and growth direction. However:

(i) at cellular scale, it is difficult to envision a mechanism that would be sensitive enough to respond to small differences in stress (this was assumed in Heisler 2010, but it remains a weak point of that study). This is however well described in animal system (membrane tension promotes exocytosis and inhibits endocytosis, see Asnacios and Hamant 2012).

(ii) mechanical stress would not explain the initial differences in growth rates in any case.

I thus agree with the authors that polarity must involve extra cues of biochemical nature, possibly working in synergy with stress. But the rationale should be better explained. In particular, in the model description "CMTs and auxin flux/concentration are the main contributors to PINs localization" is unlikely to be true since PIN localization primarily depends on actin filaments (see e.g. Geldner 2001). Rather, CMTs, as proxy of stress direction, match PIN localization. I believe that the authors would be better off with a model in which instead of cytoplasmic microtubules, actin filaments are modeled (i.e. the MFcell vector). This would not change much in terms of simulations, but it would be more accurate (actin filaments are also believed to align with tensile stress, see e.g. Goodbody and Lloyd 1990).

2) Symmetry breaking event: The idea that the radicle behaves like a trichome in sepals (in Hervieux 2017) is appealing. Maybe I would make the comparison with trichomes more obvious, because it is probably easier to grasp the idea in the case of the trichome (i.e. that a fast growing zone generates mechanical shielding (circumferential CMTs in adjacent cells) and thus anisotropic growth of the radicle). However, figure 1A,B only shows simulation, and figure 1 does not show CMT orientation. So the wording should be checked. For instance "In this scenario, we could only observe the strong isotropic growth of the root radicle without a specific orientation of CMTs (Figure 1A) » does not match with what is shown on Figure 1A. More importantly, the authors should provide evidence of CMT orientation before and during radicle outgrowth to support their claim (this could be down on fixed tissues, or with existing, even published, data). Last the authors, should also discuss growth anisotropy and growth direction (not only growth rate) to show that the CMT orientation emerges from a conflict of growth rate, and not from a conflict of growth direction (see e.g. Rebocho and Coen). This is necessary especially because the authors claim that "anisotropic growth of the root emerges from initially uniform isotropic cell expansion", hence growth anisotropy needs to be quantified. This should be an easy fix since all the relevant data are present on the video files.

3) radicle vs. root: The transition between the initial symmetry breaking event and the anisotropic growth of the root is a bit problematic to me: we are not looking at the same cells/stage anymore. It's almost as if there were two papers in one. I'm wondering whether the Heisler model could be tested at the early stages of radicle emergence, i.e. assuming that differential growth prescribes maximal membrane tension around the emerging radicle, wouldn't PIN polarize towards the radicle (assuming PIN is recruited on the most tensed membrane, because tension traps it there)? This would at least provide a link between the two parts of this study. Furthermore, this would allow the authors to test how robust (or more likely, how "unrobust") is a 100% stress-derived PIN polarity model for root growth, leading to the exploration of alternative hypothesis (phosphatase model).

4) Robustness of the model: The phosphatase model with the addition of the stem cell division rule can reproduce observed PIN patterns. This is a nice hypothesis, but without experimental support for it (i.e. PP2A phosphatases are certainly involved, and there is experimental support for that, but it is still unclear how central their role is). The simulations show that this hypothesis is plausible, but one could probably envision many other mechanisms. Thus, in absence of molecular support for the hypothesis, the central question is that of robustness. Given the number of parameters in the model, one could likely find a parameter space in which the model is robust. The robustness of the model is tested with different embryo geometry and "in a plausible range of parameter values". This part should be expanded. In particular, for which values is the model not robust anymore? The multiple tests at the end of the result section provide support for the robustness of the model. I would conclude that the model is robust only at the end of the results, and not before those tests.

*Reviewer #3 (Recommendations for the authors):*

Great study! In my opinion, it just requires some polishing and restructuring.

[Editors' note: further revisions were suggested prior to acceptance, as described below.]

Thank you for resubmitting your work entitled "A coupled mechano-biochemical model for root meristem morphogenesis" for further consideration by *eLife*. Your revised article has been evaluated by Detlef Weigel (Senior Editor) and a Reviewing Editor.

The manuscript has been improved but there are some remaining issues that need to be addressed, as outlined below:

All three reviewers are somewhat disappointed by the changes made by the authors. While a major revision was called for the authors seemed to have mostly made the easiest textual changes rather than going into certain biological details, or answered certain reviewer comments while not making corresponding changes in the text. English language and grammar are still not of high quality and at places very poor and limiting understandability.

We therefore ask for another round of revision in which you address all remaining issues, particularly the points raised with regards to biological details.

*Reviewer 1:*

Title/Abstract/Discussion

Although no claim of self-organization is made within the title, also the word morphogenesis or the statement that their model explains root shape is overselling what the authors can explain with their model. The model in its current form explains only the elongated shape of the root from the initial mechanical symmetry breaking and auxin patterning, but not its specific wedge shape nor the specific switching between division patterns close to the QC that sets up the different tissue layers. I suggest to replace morphogenesis/shape with (polarity) patterning or something in that direction. Discussion line 434/435: please remove that the model explains tissue patterning, as it does not as the authors describe themselves that they need certain rules to get the cell division patterns and hence the organization of tissue types right near the QC.

Response to comments:

– Author answer to point 27 by this reviewer: the authors should write their answer also explicitly in the discussion: that the model requires extension in terms of adding remeshing before it is suited for studying tropisms

– Author answer to point 30 raised by this reviewer: also write this explicitly in methods

– The authors now show that mechanics/AF is necessary for correct PIN and auxin patterning. First, it is not entirely clear whether mechanics were removed from these simulations (Figure 2 Suppl 7 from the start or after initially running the full model). Please clarify. Second, this nice result deserves some more attention, discussing how mechanics define the axis and PIN dynamics the polarity in the text!

*Reviewer 2:*

Rather than going into the particularities of mechanics, PIN delivery, and the roles of different parts of the cytoskeleton in this, as suggested by the reviewer, the authors have simply rephrased matters into a generic "anisotropy factor". I suggest that at least some effort into discussing the underlying biology in more depth is undertaken.

*Reviewer 3:*

It is recommended that the authors have another look at the points raised earlier and address these appropriately. As an example in their current response they indicate that simulation is a complete PIN KO (which would be embryonically lethal) yet in the paper it still states pin1 mutant. Also no appropriate response to and explicit incorporation of into manuscript of the fact that in the model columella PIN levels are predicted to be much lower than observed experimentally is given. Etc. It’s a nice model, so there is no reason to hide limitations, all models have them.

---

## [Author Response]

Essential revisions:1) The authors are required to more extensively discuss modeling details, assumptions and limitations. PDB methodology, how cytoskeleton and how PIN polarisation are modeled should get more attention in the main text. Also, the PIN patterning that is claimed to arise in a self-organized manner is clearly driven by the imposed auxin sources and sinks. Similarly, growth anisotropy derives from the growth asymmetry imposed, so it is not self-organized but has a clear directional source of information. This matters should be discussed explicitly.

We thank all three Reviewers and Editors for their hard work and valuable inputs that definitely helped to improve the manuscript. In the revised version we elaborate more on model elements as well as we explain necessity of having source or sinks as auxin has to come from somewhere (it is known to be produces in the shoot) and then the leaves the root cycling back to the shoot- all these is experimentally observed. Therefore all models include sort of boundary conditions to mimic connection to the rest of the organisms. Polarity is not driven by source or sinks but local mechanisms that act at the level of each individual cell (flux or concentration).

Sources and sinks are introduced only to connect flow from the root to the rest of the plant. However, as suggested we intended to weaken the claim of self-organization.

2) In line with the above, the authors should tone down their conclusions regarding the self-organized nature of the patterning they observe, and make explicit what inputs are needed to get the patterning they observe. This requires adjustment of the title.

See response above. Title does not tell anything about self-organization. Perhaps reviewer mentioned the abstract which has been revised accordingly. We use the word emergence when appropriate.

3) The authors should carefully check the biological correctness of the assumptions. As one reviewer pointed out the assumed relationships between microtubules, trafficking and PIN polarity are incorrect.

We carefully revised the description of assumptions to be as close as possible to observations.

Reviewer #1 (Recommendations for the authors):1. Line 47 on first mention of growth anisotropy it should be explained what exactly is meant for readers not familiar with the subject.

This issue has been amended in the revised version (Line: 46).

2. Lines 67-70: The authors couple symmetry breaking and anisotropic growth with polar auxin transport. It should be clarified to what extent the experimental evidence supports this coupling and to what extent it is an assumption.

Experimental observations suggest that growth anisotropy aligns with preferential PIN polarization in the root. As anisotropy is controlled by mechanics of underlying tissues this is likely to feedback on PIN polarity (as mechanics constrains deposition of cell wall and membrane components Heisler et al., 2010, Feraru et al. 2012, Braybrook et al., 2013). This bio-mechanical coupling is a key hypothesis addressed in this study yet it is fairly grounded based on in vivo observations.

3. Line 97 a bit more explanation of what PBD does would be nice, remains rather vague/abstract here. i.e. what do the constraints do, is there some Hamiltonian incorporating these constraints that should be minimized? Something else?

We provided expanded description of pros and cons of PBD in the revised manuscript (Lines: 96-112). The PBD constraints are sequentially projected over the vertices of the mesh by updating their position in a Gauss-Seidel-based fashion.

4. It should be clarified whether simulations shown in Figure 1A and B contain auxin dynamics (I suppose not?) or only include tissue mechanics.

This has been clarified in revised manuscript (Lines: 128-130) and figure legend has been adapted to provide explanation for principle growth directions.

Also, the authors should adjust the depiction of CMT orientation, even at 400% zoom the arrows drawn now are unfortunately hardly visible. Finally, here and for all later simulations it should be clarified in the legend for how long simulations were run.

Actually, the lines represent the principal direction of growth, we apologize that we did not specify that in the legend and we thank reviewer for spotting this issue. The PDF version send in the submission due to conversion had lower quality figures. We provide high resolution figures with revised version of the manuscript.

5. Figure 1D: on the y-axis it states "growth increase per hour", what exactly is meant here a growth rate increase, or a size increase? Please clarify.

Yes it is size increase over time. This has been clarified.

6. CTM should read CMT at line 138/146/150?

We thank reviewer for spotting this issue, it has been amended.

7. Presumably at some point also the hypocotyl growth speed will increase. Is the model capable of sustaining root growth anisotropy if after an initial period of growth rate differences between radicle and adjoining tissue the adjoining tissue starts to grow at a similar speed? How does this depend on auxin dynamics?

We tested in new (Figure 1—figure supplement 4) and root growth is robust in this scenario. Anisotropy of the organ is conserved even after the differential growth between tissues is removed.

8. Around line 145 a short mention of the relevance of intramembrane PIN diffusion and PIN internalization would be in place; anisotropic delivery of proteins only results in anisotropic patterns if diffusion is limited and/or proteins are continuously internalized and redeposited, a matter on which the last author has previously done excellent work.

Current model does not explicitly include exo- or endocytosis or lateral diffusion of proteins in the membrane. This was simplified in the current framework to reduce number of variables and unknown parameters. Revised manuscript clarifies these assumptions (Lines: 200-205).

9. Starting at lines 154 the authors compare two methods for auxin-driven PIN polarization. However, the second method, the so-called regulator polarizer mechanism, functions essentially the same as the first, the with-the-flux method, in that both promote PIN deposition at the location of highest auxin flux. The authors should explain why they compare these two PIN polarization mechanisms, what are their essential differences (i.e. why do the mechanisms give different results predominantly for lateral PIN orientations), and why they did not investigate another frequently investigated PIN polarization mechanism generally referred to as up-the-gradient polarization. The way it is presented now is confusing, as less technical readers may get the wrong impression that it is with the flux and up the gradient type mechanisms that are compared.Secondly, as the authors state, the regulator-polarizer mechanism is essentially a Turing-type reaction diffusion patterning mechanism within the cell, which typically have a characteristic parameter dependent wavelength. How does such a characteristic wavelength impact the potential of this mechanism to correctly polarize cells of different sizes as occur in the modeled root tip?

We argue that regulator-polarizer model mimics a plausible molecular flux sensor that has been so far elusive. Kinases and phosphatase affecting auxin transport were identified and could be a part of this mechanism (Lines: 432-435). We made loose comparison to reaction-diffusion model because it is locally activated and inhibited on larger distances. The wavelength here could be individual cell but in fact it is not the classical reaction-diffusion system therefore this comparison is rather loose. We therefore removed a confusing sentence from the manuscript. We discuss the different effect of the two mechanisms on PIN lateralization in Lines: 283-288. As for up-thegradient model we did not use it because it would result in incorrect polarization aligned towards auxin sources (see Author response image 1 same color coding as in main figures of manuscript).

**Author response image 1. sa2fig1:** 

10. The authors state they implement an apolar AUX/LAX pattern for cells, however it is unclear whether this pattern is applied for all cells or only for particular cell types, as has often been the case in other studies. Please clarify.

Thank you for pointing this. We have now clarified this issue (Lines: 200-205).

11. Also, in light of different modeling approaches in the field, the authors should clarify in their main text if their model allows for a single intracellular and intrawall auxin concentration or rather allows for intracellular and intrawall concentration gradients. Additionally they should clarify in the main text whether it is intra or extracellular auxin concentrations governing cell growth. Finally, if it is a single intracellular auxin concentration that governs growth, does this not imply that auxin and PIN patterning merely impact growth rate of cells and not growth anisotropy? Please clarify.

Again we thank Reviewer for spotting this confusion. We clarified these issues in the revised manuscript. Intracellular auxin drives cell growth and auxin and PINs do not regulate anisotropy (mechanics does), though they contribute to final shape of root and thus impact on mechanics (Lines: 190-195, 226-230 and 267-274).

12. The article claims to explain the self-organized growth and patterning of the plant root tip. However in the model the authors impose an auxin source in the middle vascular files and an auxin sink in the topmost outer cell files, thus quite a bit of auxin-related prepatterning is superimposed. Indeed, in a sense the polarity of auxin transport outside the modeled root tip domain was superimposed and the modeled domain should merely align accordingly, rather then truly self-organize its PIN patterning. The authors should adress this issue more explicitly.

Most plant organ models include some sort of boundary conditions to mimic connection to the rest of the organisms (Mahonen et al., 2014; Grieneisen et al., 2007, Band et al., 2012). In fact, root is connected to hypocotyl and auxin must flow in and flow out to reflect this natural phenomena. Therefore, we believe this assumption is biologically sound and necessary to retain continuity of the model section (Lines: 214222 and 497-503). Furthermore, cell polarity is not established by source or sinks but local mechanisms that act at the level of each individual cell that interpret incoming signals (i.e. flux). Without that local interpretation cells are blind as these cells do not sense global gradients; everything is locally perceived. However, as suggested by Reviewer we had weaken the claim of entirely self-organizing phenomena in the manuscript.

The authors could test whether imposing either only vascular influx or top epidermal sinks or only a transient signal would suffice to induce correct PIN patterning (I presume that in Figure 3 the auxin source is only removed after the PIN pattern was established, not from the beginning onwards) ? Even more important, the authors should test to what extent mechanical feedback is necessary for the observed PIN patterning given the imposed auxin source and sink locations and auxin-PIN feedback mechanisms (of course in these simulations PIN membrane delivery should be made independent from CMT orientation). Finally, in cells with lateral inwards PIN, is CMT organization also less longitudinally and more diagonally or even transversally oriented, and if so how does this arise from tissue growth mechanics? Please clarify.

Both components are necessary. Mechanics constrains where PINs can be deposited whereas auxin defines which side is preferred (either flux or regulator-based) and controls the rate of cell growth. When we remove mechanics feedback growth on PIN polarity is less robust (as auxin only defines growth rates and final PIN localization) , see new Figure 2 —figure supplement 7. Results of severely perturbed mechanics are presented in Figure 5E. We added clarification in the revised manuscript (Lines: 267274).

13. Line 206, please specify which type/subset of parameters could be fitted based on these data.

This has been adjusted (Lines: 238-242). PIN deposition rate was fitted to experimental data.

14. Line 210, please specify the nature of the peak (i.e. a peak of what).

It is the peak of growth rate, this has been corrected.

15. For Figure 2H-K please clarify whether the shown 1D profiles are for a specific cell file, or rather an average across all cell files for that particular position along the longitudinal axis.

It is an average across all cell files along the longitudinal axis. It has been amended in revised figure legends.

16. Line 214 please consider reformulating "eventually reproducing the non-trivial shape of the root". The model reproduces growth direction anisotropy and PIN and auxin patterning, yet as far as I can judge does not provide an explanation for the wedge shape of the root tip, or the precise organization of the SCN and the differentially oriented divisions occurring there or the precise number of cell files present in the root tip.

We have revised the manuscript to express that matter more clearly (Lines: 248-255).

17. Line 238 Please clarify the "plausible range of parameter values" statement. Over what range were parameters varied, which parameters?

The exact parameters values are provided in the legend of Figure 5—figure supplement 3. Typically were tested in range of the order of magnitude or more.

18. Line 239 please consider reformulating "prediction" into "emergent property", as the phenomenon has long been known it can hardly be seen as a model prediction, at the same time that it automatically arises from the model as an emergent property does deserve more attention.Also, the authors could possibly test the explanation they offer for the bidirectional auxin flow in the cortex by artificially manipulating epidermis/lateral root cap auxin flow and seeing how this affects cortical PIN patterning.

Adapted as suggested. We apologize for the confusion but this has been already presented in Figure 3 as we artificially removed lateralization and test its effects in combination with auxin source in QC for growth dynamics.

19. In Figure 3C it appears as if -compared to Figure 3D- PIN proteins are not or hardly present on the membrane. Please clarify and explicitly discuss this matter in the text.

In no reflux scenario we artificially manipulated auxin flow from LRC and epidermis. Therefore, in this scenario no auxin goes to the cortex this affects PIN expression levels as they depend on auxin concentrations. The visibility of PINs has been improved in high quality figures submitted in the revision.

20. Auxin levels in Figure 2C/E-G and Figure 5A in particularly the vasculature seem much lower than in Figure 3C/D. Please clarify?

This is because when the root is shorter there is more auxin, as the auxin source is always the same (in top vascular cells), the root at the beginning displays a higher concentration of auxin, and progressively loses it, we have clarified this issue in the revised manuscript (Lines: 215-222).

21. In Figure 3E and F, where in the root are growth rate or auxin concentration measured. Please clarify in figure legend.

It has been specified in Figure 3 legend in the revised version.

22. Figure 5D, the cartoon to the left suggests that in the root tip cut experiment only downward PIN mediated auxin flow occurs, please clarify if this is an emergent result from the root tip cut or rather that PIN2 mediated flow has been abolished.

It is an emergent property. This has been clarified in the Figure 5 legend.

23. Figure 5E, please clarify whether the narrowness of the root at the top is a result of imposed mechanical boundary conditions.

Yes it is attached to the rest of the plant we clarify that assumption in the revised Figure 5 legend.

24. Line 354 Please replace predict with reproduce.

Amended.

25. In the discussion the authors should be a bit more modest in their statements on the models ability to reassemble key root properties given the importance of superimposed mechanical asymmetries, auxin sources and sinks as discussed earlier.

As suggested we have weaken our claims of entire self-organization (see comments above).

26. The authors should describe in the main text in a bit more detail how exactly does auxin translate into cellular growth rate and how does this ensure stable, coordinated growth across cell files. In a previous study it was shown that since auxin levels differ significantly across cell files (e.g. much higher in vasculature than in neighboring cell files), problems in coordinated cell growth may occur (https://pubmed.ncbi.nlm.nih.gov/25358093/). Why is that not happening here. Please explain.

We thank reviewer for raising this matter. Now, we have explained what relation between auxin levels and growth rates is (Lines: 226-230). As for indicated study it has different inner working (for instance cells seems to slide against each other) and therefore it is difficult to compare with our approach. In our model cells share the common wall with their neighbors (as in real plant tissues) and therefore cell sliding is impossible. Moreover, cells are considered as almost incompressible objects (as they are filled with water). This is mimicked in our model by the internal meshing of the cell with provide resistance to compression and shearing thanks to the action of the distance constraint. This feature is described in point 1 of the suppl section: Position-based dynamics implementation. To conclude, we have never seen that problem in our simulations as biomechanics layer always compensates for overgrowing cells.

27. In the discussion authors mention possible uses of the model such as studying tropisms. However the latter requires incorporating an elongation zone in which cells undergo rapid and extreme cell elongation. It seems that the current model only incorporates slow cytoplasmic cell growth and division occurring in the meristem, would the used model formalism be capable of describing rapid cell elongation? Would the model formalism be capable of simulating the growth asymmetries occuring in root tropisms?

Although outside of scope of this study, we are exploring those possibilities in current framework. The current implementation of model will require the addition of new feature in order to allow the modelling of the elongation zone. Most notably, it would be necessary to dynamically remeshing the elongating cells in order to avoid the appearance of extremely thin and long triangles, which would definitely cause some numerical trouble to the system solution. Therefore, the extension of the current model to incorporate alternative biological processes such as rapid cell elongation are feasible but it requires the implementation of new computational procedures and thus it a matter of ongoing research.

28. The simulation code is made available to reviewers, Will the code be made publicly available upon publication?

Yes the code will be available to public upon publication.

29. Equation 4 and descriptions thereof: would it not make more sense to denote this as apoplastic rather than membrane auxin levels (although I realize apoplast and membrane are a single entity in the model formalism). Also the subscript in the first term should not read cell i but membrane/apoplast i I believe.

In fact, we formulated the amount of auxin exchanged between the intercellular space and the neighboring cells which follows the proposed description.

30. Equation 7, the authors state that auxin decay rate increases beyond a certain threshold auxin level, they should add on what experimental data this is based.

Currently, it’s just an assumption of the model intended to sustain maximal auxin levels below a certain limit (the saturation/plateau term). Although, it is known that excessive auxin levels are toxic to plants and retard their growth. Therefore it is plausible auxin homeostasis is tightly regulated to hold auxin levels in check.

31. Equations 8 and 9: what do the terms AUX1_cell*AUX1_tr and PIN_cell*PIN_tr mean? Only later clear it is about trafficking, explain at first occurrence please.

We apologize for the confusion. It is the amount of protein trafficked from the cytoplasm to the membranes; we clarify that issue in revised formulas.

32. The model contains quite a lot of non-linear interactions, with particularly for the flux and regulator-polarizer model powers of 4, the authors should clarify if model outcomes critically depend on these strong non-linearities.

Nonlinearities of higher order are used to allow thresholding effect similar to hysteresis, so polarization state can be temporally memorized. High nonlinearities assure amplification effect(increased sensitivity) which could results from multi-cascade signaling events: i.e. kinases and phosphatases such as MAPK (O'Shaughnessy et al., 2011, Cell doi.org/10.1016/j.cell.2010.12.014).

General comment: English grammar is quite poor, requires correction.

We further improved the overall readability of the manuscript.

Reviewer #2 (Recommendations for the authors):Some points would require attention:1) Misconception on the role of cytoplasmic microtubules: "CTMs restrict the deposition of various protein cargoes on the plasma membrane, typically along the maximal growth direction (maximal strain) (Adamowski et al., 2019; Nieuwland et al., 2016; Siegrist and Doe, 2007; Yang, 2008). It is plausible that PIN protein allocation (and/or other cargoes) at the plasma membrane might be restricted by CTMs." I have a problem with this claim. In Heisler 2010, it was shown that PIN1 remains polarly localized when microtubules are depolymerized. How does that fit with this model? Wouldn't it make more sense to involve the link between CESA and PIN1 (Feraru), or membrane tension and vesicle trafficking (Nakayama 2012, Heisler 2010)? I think the authors rather want to say that mechanical stress is vectorial in essence, and thus could guide both growth anisotropy and growth direction. However:(i) at cellular scale, it is difficult to envision a mechanism that would be sensitive enough to respond to small differences in stress (this was assumed in Heisler 2010, but it remains a weak point of that study). This is however well described in animal system (membrane tension promotes exocytosis and inhibits endocytosis, see Asnacios and Hamant 2012).(ii) mechanical stress would not explain the initial differences in growth rates in any case.I thus agree with the authors that polarity must involve extra cues of biochemical nature, possibly working in synergy with stress. But the rationale should be better explained. In particular, in the model description "CMTs and auxin flux/concentration are the main contributors to PINs localization" is unlikely to be true since PIN localization primarily depends on actin filaments (see e.g. Geldner 2001). Rather, CMTs, as proxy of stress direction, match PIN localization. I believe that the authors would be better off with a model in which instead of cytoplasmic microtubules, actin filaments are modeled (i.e. the MFcell vector). This would not change much in terms of simulations, but it would be more accurate (actin filaments are also believed to align with tensile stress, see e.g. Goodbody and Lloyd 1990).

Yes we agree it is quite more complicated how PIN dynamics is regulated. Indeed neither CMTs nor Actin filaments inhibitors alone does not tend to perturb PIN localization dramatically. To open possibilities for either, we have now introduced the anisotropy factor (AF) along the manuscript which combines the action of CMTs, actin and cell wall component deposition for clarification as description of mechanical impact on PIN polarization (Lines: 166-172, 571-580).

2) Symmetry breaking event: The idea that the radicle behaves like a trichome in sepals (in Hervieux 2017) is appealing. Maybe I would make the comparison with trichomes more obvious, because it is probably easier to grasp the idea in the case of the trichome (i.e. that a fast growing zone generates mechanical shielding (circumferential CMTs in adjacent cells) and thus anisotropic growth of the radicle). However, figure 1A,B only shows simulation, and figure 1 does not show CMT orientation. So the wording should be checked. For instance "In this scenario, we could only observe the strong isotropic growth of the root radicle without a specific orientation of CMTs (Figure 1A) » does not match with what is shown on Figure 1A.

We thank Reviewer for rising this issue. We follow the suggestion and include indication of similarity to trichome (Lines: 114-118). We does not show microtubule orientation but principal growth direction instead (clarified in figure legends). As suggested we refer now to anisotropy factor (which could be combine action of CMTs, actin and cell wall components) that impacts on PIN polarization and growth mechanics.

More importantly, the authors should provide evidence of CMT orientation before and during radicle outgrowth to support their claim (this could be down on fixed tissues, or with existing, even published, data).

This mention been included in the revised manuscript (Lines: 120-126).

Last the authors, should also discuss growth anisotropy and growth direction (not only growth rate) to show that the CMT orientation emerges from a conflict of growth rate, and not from a conflict of growth direction (see e.g. Rebocho and Coen). This is necessary especially because the authors claim that "anisotropic growth of the root emerges from initially uniform isotropic cell expansion", hence growth anisotropy needs to be quantified. This should be an easy fix since all the relevant data are present on the video files.

To demonstrate further that it is growth rates rather than conflicts that drive symmetry breaking, we run additional simulations and results are presented in new Figure 1— figure supplement 3 and 4 (Lines: 143-146).

3) radicle vs. root: The transition between the initial symmetry breaking event and the anisotropic growth of the root is a bit problematic to me: we are not looking at the same cells/stage anymore. It's almost as if there were two papers in one. I'm wondering whether the Heisler model could be tested at the early stages of radicle emergence, i.e. assuming that differential growth prescribes maximal membrane tension around the emerging radicle, wouldn't PIN polarize towards the radicle (assuming PIN is recruited on the most tensed membrane, because tension traps it there)? This would at least provide a link between the two parts of this study. Furthermore, this would allow the authors to test how robust (or more likely, how "unrobust") is a 100% stress-derived PIN polarity model for root growth, leading to the exploration of alternative hypothesis (phosphatase model).

We are unsure what the reviewer meant by two different papers. We simulate radicle emergence from “late embryonic stage”. We provide continuous simulations where mechanics only restricts PINs to the axis of growth but does not tell where PINs will be finally deposited this is done through flux sensing. Therefore, it is very unlikely up-thegradient model would polarize PINS away from auxin source (see simulation presented in Author response image 2 (color coding as in main figures of manuscript)). We clarify this issue in revised manuscript (Lines: 166-172).

4) Robustness of the model: The phosphatase model with the addition of the stem cell division rule can reproduce observed PIN patterns. This is a nice hypothesis, but without experimental support for it (i.e. PP2A phosphatases are certainly involved, and there is experimental support for that, but it is still unclear how central their role is). The simulations show that this hypothesis is plausible, but one could probably envision many other mechanisms. Thus, in absence of molecular support for the hypothesis, the central question is that of robustness. Given the number of parameters in the model, one could likely find a parameter space in which the model is robust. The robustness of the model is tested with different embryo geometry and "in a plausible range of parameter values". This part should be expanded. In particular, for which values is the model not robust anymore? The multiple tests at the end of the result section provide support for the robustness of the model. I would conclude that the model is robust only at the end of the results, and not before those tests.

In revised version we moved the summary of robustness to the last part of the manuscript (Lines: 400-406) and exact parameter values tested are presented in figure legends (Figure 5 – figure suppl. 3). Since this model is already quite complex we had to adapt the PBD framework to run on supercomputing cluster to generate results in this manuscript which was a non-trivial and difficult task.

Reviewer #3 (Recommendations for the authors):Great study! In my opinion, it just requires some polishing and restructuring.

We thank Reviewer for showing enthusiasm and again for all suggestions that helped in improving this manuscript.

[Editors' note: further revisions were suggested prior to acceptance, as described below.]

We therefore ask for another round of revision in which you address all remaining issues, particularly the points raised with regards to biological details.Reviewer 1:Title/Abstract/DiscussionAlthough no claim of self-organization is made within the title, also the word morphogenesis or the statement that their model explains root shape is overselling what the authors can explain with their model. The model in its current form explains only the elongated shape of the root from the initial mechanical symmetry breaking and auxin patterning, but not its specific wedge shape nor the specific switching between division patterns close to the QC that sets up the different tissue layers. I suggest to replace morphogenesis/shape with (polarity) patterning or something in that direction.

We thank the reviewer for these suggestions. To address these issues, we did the following:

– The title was modified to “A coupled mechano-biochemical model for cell polarity guided anisotropic root growth”.

– The abstract has been modified by substituting “morphogenesis” with a more generic “growth”, and “polar patterning”.

– We clearly distinguish between emergent properties of the model (cell polarity, anisotropy, PIN localization, and auxin distribution) and model assumptions/simplifications. Some major changes were done to the manuscript and method sections to further clarify any misunderstanding and confusion (i.e., see Discussion, Method sections, abstract (Lines 25-28), Line 72, Lines 104106, 112-124, 142-151, 163-180, 207-221, 234-238, 250-253, 280-290, among others).

We discuss model limitations and suggest further work to improve these bottlenecks (see revised Discussion).

Discussion line 434/435: please remove that the model explains tissue patterning, as it does not as the authors describe themselves that they need certain rules to get the cell division patterns and hence the organization of tissue types right near the QC.

Amended, Lines 443-447.

Response to comments:– Author answer to point 27 by this reviewer: the authors should write their answer also explicitly in the discussion: that the model requires extension in terms of adding remeshing before it is suited for studying tropisms.

This has been explained in detail in the revised Discussion, Lines 468-476, together with a description of the model’s limitations and shortcomings.

– Author answer to point 30 raised by this reviewer: also write this explicitly in methods.

This has been explained in the Method section Auxin transport module description Lines 722-726.

– The authors now show that mechanics/AF is necessary for correct PIN and auxin patterning. First, it is not entirely clear whether mechanics were removed from these simulations (Figure 2 Suppl 7 from the start or after initially running the full model). Please clarify. Second, this nice result deserves some more attention, discussing how mechanics define the axis and PIN dynamics the polarity in the text!

We thank the reviewer for pointing this out. This AF effect on PIN trafficking was removed at the start of each simulation. We also provide discussion on that matter, suggesting an important role of growth mechanics in cell polarity maintenance see Lines 280-290 and Figure 2—figure supplement 7 legend.

Reviewer 2:Rather than going into the particularities of mechanics, PIN delivery, and the roles of different parts of the cytoskeleton in this, as suggested by the reviewer, the authors have simply rephrased matters into a generic "anisotropy factor". I suggest that at least some effort into discussing the underlying biology in more depth is undertaken.

We thank the reviewer for this suggestion.

In the substantially revised manuscript and methods, we provide the reasoning underlying AF introduction as the output of still poorly understood cytoskeleton shaping processes (Lines 112-124) and we also present a scheme to explain how AF feeds back on growth mechanics for clarification (see Method section Position-based dynamics implementation, Lines 582-596). We made this simplification to reduce already complex models, otherwise, we would just increase the number of unknown parameters. When appropriate we point out model simplifications and suggest further expansion of the model once more experimental data become available (see i.e., Discussion Lines 468-476).

Similarly, we tend to avoid overcomplication of the model by including specific processes leading to PIN trafficking such as endo and exocytosis membrane diffusion or CMT/actin dynamics (see explanation Lines 163-180, and 207-221). The general idea is to demonstrate minimal design principles that lead to cell polarization, anisotropic growth, and auxin transport patterning.

Reviewer 3:It is recommended that the authors have another look at the points raised earlier and address these appropriately. As an example in their current response they indicate that simulation is a complete PIN KO (which would be embryonically lethal) yet in the paper it still states pin1 mutant. Also no appropriate response to and explicit incorporation of into manuscript of the fact that in the model columella PIN levels are predicted to be much lower than observed experimentally is given. Etc. It s a nice model, so there is no reason to hide limitations, all models have them.

We thank the reviewer for pointing this issue. In the revised version, we clarified assumptions and simplifications that we made in our models, also presenting a plan to improve them in the next versions of these models (See Discussion). Major changes have been made to the overall manuscript and Method sections which we believe address all remaining issues.

As suggested, we point out the difference between model predictions and experiments regarding columella and explain why we observe these differences (see Lines 214221).

Results of model simulations are presented as they are and open for possible alternative interpretations. Furthermore, method sections have been improved to increase further the clarity. We do point out assumptions and shortcomings of our models and leave a space for further model improvement. We also must agree no models are perfect – there are and will be a simplification of real complexities that we observe in nature. Yet we believe models as the ones presented here will be very useful to understand the principles behind nature’s complexity and more importantly make predictions to aid in experimental design.

Regarding the PIN1 issue, we replace notation for more generic “vascular PINs” see (Lines 373-377 and Figure 5—figure supplement 1A). It is important to mention that all simulated mutants reflect the reduction of PINs (knockdown) as opposed to full knockouts. Nevertheless, if this issue remains confusing, we are open to remove “vascular PIN” simulations from the manuscript.